# New insights into the role of microheterogeneity of ZP3 during structural maturation of the avian equivalent of mammalian zona pellucida

Hiroki Okumura[1]*, Ayaka Mizuno[1], Eri Iwamoto[1], Rio Sakuma[1], Shunsuke Nishio[2], Ken-ichi Nishijima[3], Tsukasa Matsuda[4], Minoru Ujita[1]

1 Department of Applied Biological Chemistry, Faculty of Agriculture, Meijo University, Nagoya, Japan,
2 Department of Biosciences and Nutrition, Karolinska Institutet, Huddinge, Sweden, 3 Avian Bioscience Research Center, Graduate School of Bioagricultural Sciences, Nagoya University, Tokai National Higher Education and Research System, Nagoya, Japan, 4 Faculty of Food and Agricultural Sciences, Fukushima University, Fukushima, Japan

* hokumura@meiju-u.ac.jp

## Abstract

The egg coat including mammalian zona pellucida (ZP) and the avian equivalent, i.e., inner-perivitelline layer (IPVL), is a specialized extracellular matrix being composed of the ZP glycoproteins and surrounds both pre-ovulatory oocytes and ovulated egg cells in vertebrates. The egg coat is well known for its potential importance in both the reproduction and early development, although the underlying molecular mechanisms remain to be fully elucidated. Interestingly, ZP3, one of the ZP-glycoprotein family members forming scaffolds of the egg-coat matrices with other ZP glycoproteins, exhibits extreme but distinctive microheterogeneity to form a large number of isoelectric-point isoforms at least in the chicken IPVL. In the present study, we performed three-dimensional confocal imaging and two-dimensional poly-acrylamide-gel electrophoresis (2D-PAGE) of chicken IPVLs that were isolated from the ovarian follicles at different growth stages before ovulation. The results suggest that the relative proportions of the ZP3 isoforms are differentially altered during the structural maturation of the egg-coat matrices. Furthermore, tandem mass spectrometry (MS/MS) analyses and ZP1 binding assays against separated ZP3 isoforms demonstrated that each ZP3 isoform contains characteristic modifications, and there are large differences among ZP3 isoforms in the ZP1 binding affinities. These results suggest that the microheterogeneity of chicken ZP3 might be regulated to be associated with the formation of egg-coat matrices during the structural maturation of chicken IPVL. Our findings may provide new insights into molecular mechanisms of egg-coat assembly processes.

**Data Availability Statement:** All relevant data are within the paper and its Supporting Information files.

**Funding:** This work is financially supported in part by Grants-in-Aid for Scientific Research (C) [Grant Number 25450520] from the Ministry of Education, Culture, Sports, Science, and Technology of Japan, and by Grants for Encouragement of Scientific Research from the Research Institute of Meijo University. The funders had no role in study design, data collection and analysis, decision to publish, or preparation of the manuscript.

## Introduction

The zona pellucida (ZP), a mammalian egg coat, is a specialized extracellular matrix that is formed between the growing oocyte and the surrounding ovarian follicle cells (i.e., granulosa or cumulus cells) during the follicle development in the ovary [1]. Notably, the ZP is assembled exclusively from the ZP glycoproteins that are classified into at least 6 subfamilies (ZP1, ZP2, ZP3, ZP4, ZPD, and ZPAX) based on phylogenetic analyses [2, 3]. The ZP enveloping an ovulated egg is well known to be involved in the sperm-egg interactions, including the induction of sperm acrosome reaction and the polyspermy blocking immediately before fertilization [4–6], and in the normal traversing of pre-implantation embryos through the oviduct into the uterus until hatching [7]. However, interestingly, it has been reported that the ovarian follicle of the ZP-glycoprotein knock-out female mice ($mZP1^{-/-}$, $mZP2^{-/-}$, and $mZP3^{-/-}$) exhibited less orderly arranged follicle cells than that of the wild-type ones, in addition to the loss or malformation of ZP [8]. Furthermore, it is suggested in some cases of human female infertility that mutations in the ZP1, ZP2, or ZP3 genes are associated with the oocyte degeneration in the ovarian follicles of patients [9–12]. These findings imply that the ZP and ZP glycoproteins play significant roles not only in the sperm–egg interactions and embryogenesis, but also in the histogenesis of ovarian follicle, or supporting communications between the oocyte and the ovarian follicle cells during follicular development [8]. Therefore, understanding the mechanisms of egg-coat formation could provide the basis for developing new approaches to control human and animal fertility, including new therapies for infertility that might be related to the egg-coat abnormality.

Despite the importance of ZP in female reproduction, the underlying molecular mechanisms are poorly understood, especially in the regulation of ZP-matrix assembly from ZP glycoproteins. To elucidate these mechanisms, it might be necessary to characterize the structural, biochemical, and physiological properties of ZP and ZP glycoproteins throughout the ZP-assembly processes using the ZP samples collected from the ovarian follicles of different growth stages. Here, we focused on the avian equivalent of mammalian ZP, i.e., the inner-perivitelline layer (IPVL) [13], as a model for studying the ZP-matrix formation. This is because the avian ovarian follicles and oocytes generally grow much larger than mammalian ones before ovulation [14], and therefore, vast amounts of IPVL at different stages of the formation process can easily be isolated from the macroscopically distinguishable follicles [15, 16]. Especially in poultry species, including chicken and quail etc., ovulation occurs approximately once every day in sexually mature females, and therefore, the IPVL surrounding their rapidly growing oocytes are formed quickly and might be suitable for investigation of the egg-coat matrix assembly processes.

In a chicken ovary, the primordial follicles containing the primary oocytes develop into the primary follicles (~1 to ~2 mm in diameter) during the sexual maturation, and subsequently, the primary follicles develop into the white follicles (~6 to ~8 mm in diameter) coordinately with the uptake of protein-rich white yolk into the primary oocyte until 8 to 9 days before ovulation [14]. Thereafter, the sequentially selected white follicles (in approximately one-day cycles) develop into the yellow follicles (~8 to ~40 mm in diameter) that grow rapidly with the accumulation of a large amount of lipid-rich yellow yolk in the oocytes to mature and be ovulated [17]. The yellow follicles in an individual chicken ovary are named as F1, F2, F3, and so on, in the order of their sizes and growth stages, or the order in which the oocyte will be ovulated [15, 16]. That is, the F1 is the largest and the most mature yellow follicle that will ovulate within about 1 day, the F2 is the second largest but immature one that will ovulate in the next ovulation cycle, the F3 is the third largest and more immature one that will ovulate within about 1 day after the ovulation of F2, and so on. Our previous study suggested that a probable

core structure of the chicken egg-coat matrix in the IPVL of the most mature yellow follicle F1 is composed of the insoluble ZP1–ZP3 fibers [18]. In chicken, the ZP3 gene is expressed predominantly in the ovarian granulosa cells, and its expression levels are significantly up-regulated through the stages of yellow-follicle development [19], while ZP1 gene is highly expressed in the liver cells of sexually mature laying hen [20] probably in a folliculogenesis-independent manner. Considering that the aforementioned ZP1–ZP3 fibers are thought to be self-assembled from ZP1 and ZP3 that are individually secreted from the liver cells into the blood circulation and from the granulosa cells into the extracellular space of the oocyte [21–23], respectively, it might be for these reasons that protein contents of ZP1 and ZP3 increase synchronously in the chicken IPVL of growing yellow follicles [24]. In addition, at least in chicken, a mature IPVL surrounding the oocyte just before ovulation individually contains a series of isoelectric-point isoforms of ZP3 [25]. Although it is suggested that such charge microheterogeneities of proteins are associated with the post-translational modifications [26, 27], the regulatory mechanisms and the functional significances of the microheterogeneity of ZP3 remain to be explored.

In the present study, we aimed to investigate the process of egg-coat matrix formation from ZP glycoproteins using the chicken model. For this purpose, we initially collected chicken IPVLs from the yellow follicles of the middle to late growth stages (F3, F2, and F1, respectively) to obtain three-dimensional (3D) images of their egg-coat matrix network by using confocal immunofluorescence microscopy and to determine the compositions of the ZP3 isoforms in individual IPVLs at the different maturities by using two-dimensional polyacrylamide gel electrophoresis (2D-PAGE). These experiments showed that the immature IPVLs with loose meshwork rapidly develop into dense layers of 3D matrices in a few days, and the proportions of major five to six ZP3 isoforms in each IPVL are altered significantly (P = 0.0067, by chi-square test) through the middle to late growth stages of yellow follicles. Furthermore, we estimated differences in the modifications of the major ZP3 isoforms by using tandem mass spectrometry (MS/MS), and compared relative binding affinities between ZP1 and the ZP3 isoforms that were separated from the mature IPVLs by gel-filtration chromatography and liquid-phase isoelectric focusing (LP-IEF), respectively. These experiments suggested, surprisingly, that the major ZP3 isoforms of mature IPVL exhibit distinctive patterns of not only post-translational modifications but also amino-acid substitutions at specific residues, and that there are apparent differences in the ZP1-binding affinities among ZP3 isoforms. To our knowledge, this is the first report providing insights into the structural maturation process of the egg-coat matrix and its relationship with the alterations of the ZP glycoprotein components during ovarian follicle development. Our results will be useful for understanding the egg-coat formation mechanisms.

## Materials and methods

All animal procedures were performed according to the guidelines for the care and use of experimental animals of Meijo University and approved by an institutional animal care and use committee (2015-A-E-2). Fresh blood and ovary of the commercial White-Leghorn hens were obtained from a local chicken slaughter plant. In this plant, large number of White-Leghorn hens (~15,000 hens/day) are killed by bleeding from carotid artery using a mechanical knife in an automated line system and slaughtered by hand, followed by a brief scalding in ~63˚C of hot water for 80 seconds to remove feathers. The blood and ovary were collected during the bleeding and immediately after the removal of feathers, respectively, and transported to our laboratory in an ice box. To analyze the genetically homogeneous line of hen, fresh ovary and genomic DNA of the long-term closed colony of White-Leghorn breed (WL-G)

hens that were killed similarly by hands were provided by the Nagoya University Avian Bioscience Research Center (ABRC) through the National Bio-Resource Project of the Ministry of Education, Culture, Sports, Science and Technology (MEXT), Japan. All recombinant DNA experiments were performed according to the guidelines of the Meijo University safety committee for recombinant DNA experiments with approval (2011–2201 and 2016–2201).

## Preparation of chicken serum and IPVL

Chicken serum was prepared as described previously [22]. The aforementioned fresh blood of laying hen was coagulated overnight at 4˚C, and centrifuged at 1000 x g for 30 min to remove a clot. The supernatant was collected as the laying-hen serum and stored at –20˚C. Chicken IPVL was isolated mechanically with forceps from the pre-ovulatory yellow follicles of the aforementioned ovaries of the commercial White-Leghorn and the WL-G hens as described previously [18], and stored at –20˚C. Respective, approximately 30 of yellow follicles at each growth stage (F1–F3) were collected from 3 kg of the ovaries of the commercial White-Leghorn hens, and the pieces of IPVLs being isolated from the ~30 ovarian follicles were randomly selected and subjected to each experiment. On the other hand, 2 ovarian follicles for each growth stages were collected from 2 of the WL-G hens, and pieces of the IPVLs being isolated from one of these follicles were subjected to each experiment.

## Cloning of ZP3 gene from the genomic DNA of WL-G chicken

PCR was performed using LA Taq$^{®}$ DNA polymerase (Takara Bio, Tokyo, Japan) for the genomic region of WL-G chicken containing complete CDS of the ZP3 gene with primers that were designed to amplify the region from 3088060 to 3090874 of the *Gallus gallus* breed Red Jungle Fowl isolate RJF #256 chromosome 10, GRCg6a (GenBank accession no. NC_006097.5). The designed primer sequences are as follows: Forward primer, 5′–**CGGGAATT**CCAAAGAGGCTGACGAG–3′, and Reverse primer, 5′–**ATCGAATTC**TAACACGTGGTGACCAGA–3′ (the 5′ tails being not complementary to the target sequences are shown in bold letters, and *Eco*RI restriction sites being introduced for future use are underlined). The PCR products were cloned into the pGEM-T easy vector (Promega, Madison, WI) by the TA cloning and sequenced using Applied Biosystems™ BigDye™ Terminator v3.1 Cycle Sequencing Kit (Thermo Fisher Scientific, Carlsbad, CA), both according to the manufacturer's instructions.

## Immunofluorescent staining and confocal laser-scanning microscopy of chicken IPVL

Immunofluorescent staining of chicken IPVL was performed as described previously [28]. Small pieces of the aforementioned IPVLs isolated from the ovary of commercial White-Leghorn hens were mounted onto aminosilane-coated glass slides after the treatment with 0.1% Triton X-100 without any fixation. Herein, the pieces of IPVLs were partially immobilized onto the glass slides with gentle twisting in order to expose both surfaces of them to the blocking agent and antibodies. After blocking with BSA, the IPVLs were stained with mouse polyclonal antiserum against recombinant chicken ZP1 repeat (avian Pro/Thr/Leu-rich) region or trefoil domain (anti-ZP1 repeat or trefoil, respectively) [28], which both the recombinant domains were derived from our previously cloned cDNA of chicken ZP1 (GenBank accession no. LC128587), followed by incubation with the Molecular Probes™ Alexa Fluor$^{®}$ 488 goat anti-mouse IgG (Thermo Fisher Scientific, Carlsbad, CA). Imaging was performed on a Zeiss Axio Observer.Z microscope equipped with LSM 700 laser scanning confocal optics (Carl Zeiss, Thornwood, NY). A 488-nm diode laser excitation in combination with a 490–635 nm band-pass emission filter was used for imaging the fluorescent probes. Confocal images were

taken at 0.40 or 0.68 μm intervals along the z-axis. Differential-interference-contrast (DIC) microscopy images were taken on the same system.

## Gel electrophoresis and immunoblotting of chicken IPVL

SDS-PAGE was performed according to the method of Laemmli [29]. Protein samples were incubated at 100˚C for 5 min in SDS-PAGE sample buffer (62.5 mM Tris-HCl, pH6.8, containing 2.0% SDS, 0.0020% bromophenol blue, 10% glycerol) in the absence of any reducing agents (non-reducing conditions) and loaded onto the polyacrylamide gel following removal of the insoluble materials. 2D-PAGE was performed as described previously [25] using ZOOM® IPGRunner™ System (Thermo Fisher Scientific, Carlsbad, CA). Invitrogen™ ZOOM™ Strip pH3–10NL (Thermo Fisher Scientific) was rehydrated in 150 μl of sample rehydration buffer (9 M Urea, 2% (w/v) CHAPS, 0.5% (v/v) Invitrogen™ ZOOM™ Carrier Ampholyte pH3–10 (Thermo Fisher Scientific), 0.002% bromophenol blue) containing IPVL that was collected from the F1 follicle of the WL-G hen (180 μg wet weight) or the commercial White-Leghorn hen (89 μg wet weight) for the following mass spectrometry, IPVL that was collected from the F1, F2 or F3 follicle of the commercial White-Leghorn hen (each 88 μg wet weight), or the below-mentioned fractions from the liquid-phase isoelectric focusing (LP-IEF) of ZP3 isoforms (each 10 μl). Urea in the rehydration buffer was deionized using AG 501-X8(D) Resin (Bio-Rad, Hercules, CA) before use. Both the 1st dimensional isoelectric focusing (IEF) and the 2nd dimensional SDS-PAGE were performed in non-reducing conditions. Carbonic Anhydrase Isozyme I and II from human and bovine erythrocytes, respectively (Merck, Darmstadt, Germany), were used for the IEF markers. Silver staining, Coomassie Brilliant Blue (CBB) staining, and immunoblotting were performed as described previously [18]. Silver stain MS Kit (Fujifilm Wako Pure Chemical Corporation, Osaka, Japan) was used for both the staining and de-staining of the gels in which the protein samples being subjected to the following mass spectrometry were contained. For immunoblotting, the above-mentioned anti-ZP1 repeat and mouse polyclonal antisera against recombinant chicken ZP3 ZP-C domain (anti-ZP3) [28] were used for the primary antibodies, and peroxidase-linked anti-mouse IgG (Mouse IgG HRP Linked Whole Ab (from Sheep); Merck) were used for the secondary antibody. The signal detection was performed using Cytiva Amersham™ ECL™ Western Blotting Detection Reagents (Cytiva, Marlborough, MA). To remove antibodies from the nitrocellulose membrane for reprobing, Thermo Scientific™ Restore™ Western Blot Stripping Buffer (Thermo Fisher Scientific) was used according to the manufacturer's instructions.

## Tandem mass spectrometry analyses of isolated chicken ZP3 isoforms

In-gel digestion of ZP3 isoforms were performed as follows: The spots of ZP3 isoforms that were detected on the 2-D gels were excised, washed with the washing solution containing 30% acetonitrile and 100 mM $NH_4HCO_3$ to be fully destained, and dried using Thermo Scientific™ Savant™ SpeedVac™ (Thermo Fisher Scientific, Carlsbad, CA). Proteins in the dried gels were reduced and alkylated by incubation in 10 mM dithiothreitol (DTT) for 15 min at 37˚C and subsequently in 1% acrylamide for 15 min at room temperature. After similar washing and drying, the gels were incubated in the protease solution containing Trypsin/Lys-C Mix, Mass Spec Grade (Promega, Madison, WC), 0.01% Protease MAX surfactant (Promega), and 50 mM $NH_4HCO_3$ for 60 min at 50˚C according to the manufacturer's instructions for in-gel digestion of the proteins. The resultant peptide solutions were filtrated using Ultrafree®-MC centrifugal filter, 0.1 μm (Merck, Darmstadt, Germany) to remove insoluble aggregates.

Liquid chromatography-matrix-assisted laser desorption/ionization tandem mass spectrometry (LC-MALDI MS/MS) of the tryptic peptides was performed as follows: The peptides

derived from each ZP3 isoform were separated by reverse-phase chromatography and spotted onto the μFocus MALDI plate (Hudson Surface Technology, West New York, NJ) using the nanoLC system DiNa (Techno Alpha, Tokyo, Japan) with α-cyano-4-hydroxycinnamic acid (α-CHCA) as a MALDI matrix. MS/MS spectra were collected on TOF/TOF™ 5800 System (SCIEX, Framingham, MA). Data analyses were performed using the Mascot Server (http://www.matrixscience.com/search_form_select.html; Matrix Science, Boston, MA).

The "Mascot MS/MS ions search" using the "NCBIprot database" (Taxonomy: "Other lobe-finned fish and tetrapod clade") was performed against the MS/MS datasets of each ZP3 isoform initially without the "Error tolerant search". The "Peptide mass tolerance" and the "Fragment mass tolerance" were set to ±0.1 Da and ±0.3 Da, respectively. Subsequently, the "Manual error tolerant search" was then performed for the ZP3 of the commercial White-Leghorn chicken with the "Peptide mass tolerance" of ±0.1 Da and the "Fragment mass tolerance" of ±0.1 or ±0.2 Da (the lowest absolute values of the tolerance on the order of 0.1 that were necessary to detect at least one peptide containing modifications), and for the ZP3 of WL-G chicken with the Peptide mass tolerance of ±0.1 Da and the Fragment mass tolerance of ±0.4, ±0.5, ±1.0, or ±1.1 Da (the lowest absolute values of the tolerance on the order of 0.1 that were necessary to detect both the A43 and V43 heterozygous mutations being derived from the aforementioned ZP3 gene of WL-G chicken as peptides containing the A43V and/or V43A amino-acid substitutions, each other). The "Mascot ions scores" for the detected peptides and sequences were automatically calculated from the significance levels of the detections. The Unimod database (http://www.unimod.org/modifications_list.php) was used to classify the detected modifications.

## Preparative liquid-phase isoelectric focusing of chicken ZP3 isoforms

Fractionation of ZP3 isoforms was performed using MicroRotofor™ Liquid-Phase IEF Cell (Bio-Rad, Hercules, CA) according to the manufacturer's instructions. The chicken IPVL (33 or 29 μg of wet weight) from which the ZPD was dissociated and removed by sonication and centrifugation [18] was solubilized in 8% carrier ampholyte (Bio-Lyte® 3/10 or 5/7 Ampholytes for pH range of ~3.5–9.5 or ~5–7, respectively; Bio-Rad) containing 3% Triton X-100 and 8M urea (ultra-pure grade; MP Biomedicals, Irvine, CA) and subjected to the liquid-phase isoelectric focusing (LP-IEF). The electrolyte solutions for the anode and cathode were 0.1M $H_3PO_4$ and 0.1M NaOH in combination with the aforementioned Bio-Lyte® 3/10, and 0.1M glutamate and 0.1M ethanolamine in combination with the aforementioned Bio-Lyte 5/7, respectively. Each 5 μl of the collected fractions were subjected to SDS-PAGE analysis, and the remaining fractions containing ZP3 were diluted in 3% Triton X-100 and 6M urea (ultra-pure grade; MP Biomedicals) and further subjected to the re-fractionation to separate ZP3 isoforms with higher resolution. After the re-fractionation, each 5 μl of the collected fractions were similarly analyzed by SDS-PAGE, and the rest of fractions were adjusted to 1M NaCl to strip the ampholytes from the proteins, and dialyzed against phosphate-buffered saline (PBS; 8.1 mM $Na_2HPO_4$, 1.4 mM $KH_2PO_4$, 2.7 mM KCl, 137 mM NaCl). Each 10 μl of the dialyzed fractions were subjected to 2D-PAGE to examine how the ZP3 isoforms were separated, and the rest of fractions were stored at –20˚C. ZP3 spots detected in the 2D-PAGE gels were assigned to each ZP3 isoforms based on the aforementioned IEF markers and/or comparison of the spot patterns to that of ZP3 isoforms in the F1 IPVL.

## Examination of the binding capacity of chicken ZP1 in the serum for nickel-affinity magnetic beads

PureProteome™ Nickel Magnetic Beads (30 μl of the 75–80 wt% suspension; Merck, Darmstadt, Germany) were equilibrated in the wash buffer (50 mM phosphate buffer, pH8.0,

containing 300 mM NaCl) before use. The equilibrated beads were re-suspended in 150 μl of the laying-hen serum that was diluted with 150 μl of phosphate-buffered saline (PBS; 8.1 mM $Na_2HPO_4$, 1.4 mM $KH_2PO_4$, 2.7 mM KCl, 137 mM NaCl), incubated for 16 hours at 4°C with gentle agitation, pulled by the magnetic field, and washed four times with 150 μl of the wash buffer. After washing, the beads were re-suspended in 150 μl of the elution buffer (PBS containing 300 mM NaCl, 300 mM imidazole and 0.2% Triton X-100) and pulled by a magnetic field to collect the supernatant containing proteins that were eluted from the beads. The elution process was repeated three times for the same beads. Laying-hen serum (0.2 μl for silver staining or 1.0 μl for immunoblotting), the flow-through (0.4 μl for silver staining or 2.0 μl for immunoblotting), and each the wash and elution fractions (5.0 μl for both silver staining and immunoblotting) were subjected to SDS-PAGE.

## ZP1–ZP3 binding assay

The chicken IPVL that was solubilized using 6M urea (ultra-pure grade; MP Biomedicals, Irvine, CA) was subjected to gel filtration chromatography, and the fraction containing both monomer and disulfide-linked dimer of ZP1 was obtained as described previously [18]. ÄKTA prime plus (Cytiva, Marlborough, MA) equipped with HiPrep™ 16/60 Sephacryl™ S-400 HR column (Cytiva) was used for the chromatography, and the fraction was dialyzed against PBS to be used as the ZP1 solution. The soluble $His_6$-tagged thioredoxin was expressed in *Escherichia coli* BL21(DE3) (Merck, Darmstadt, Germany), being transformed with the empty pET-32a(+) vector (Merck) and purified as described previously [28] to be used as the blocking agent. The aforementioned ZP3-isoform containing fractions from the LP-IEF were diluted with phosphate-buffered saline (PBS; 8.1 mM $Na_2HPO_4$, 1.4 mM $KH_2PO_4$, 2.7 mM KCl, 137 mM NaCl) at dilution factors of ~3–11 to nearly equalize the signal intensities of ZP3 in the immunoblotting to be used as the crude ZP3-isoform fractions with the addition of NaCl up to 300 mM.

PureProteome™ Nickel Magnetic Beads (30 μl of suspension containing 75–80 wt% beads; Merck) were equilibrated as described above, re-suspended in 150 μl of the aforementioned ZP1 solution, incubated for 16 hours at 4°C with gentle agitation, and pulled by magnetic field. After washing with the wash buffer (50 mM phosphate buffer, pH8.0, containing 300 mM NaCl), the beads were re-suspended in 150 μl of 1.0 μg/μl solution of the above-mentioned blocking agent to block an excess of nickel ions on the beads, incubated for 16 hours at 4°C with gentle agitation, and pulled by a magnetic field. The blocked beads were washed similarly, re-suspended in each 150 μl of the above-mentioned crude ZP3-isoform fraction, incubated for 16 hours at 4°C with gentle agitation, and pulled by a magnetic field. Then after washing with the wash buffer containing 0.04% Triton X-100 three times, proteins on the beads were eluted with 200 μl of the above-mentioned SDS-PAGE sample buffer by incubation at 100°C for 5 min, and the beads were pulled by a magnetic field to be removed. Finally, the crude ZP3-isoform fractions (each 15 μl) and the corresponding elution fractions (each 20 μl) were subjected to SDS-PAGE.

## Image processing and analyses

The 3D volume and surface renderings of the confocal sections of chicken IPVLs were performed using the ImageJ 1.47n software (http://imagej.nih.gov/ij/) with the 3D Viewer plugin (Plugins > 3D > 3D Viewer; with the Threshold settings of 0 and the Resampling-factor settings of 1 for the 3D volume renderings, and with the Threshold settings of 35–50 and the Resampling-factor settings of 1 for the 3D surface renderings). Prior to the 3D renderings, datasets of the confocal images were smoothed (Plugins > Process > Smooth (3D); with the

Method settings of Gaussian and the Sigma settings of 0.001) and re-sliced vertically at intervals of 0.1 μm to improve the depth resolution (Image > Stacks > Reslice), using the same software. The diameters of the surface-exposed fibers on IPVLs were measured at a randomly selected point on each 10 of the single fiber that connects the randomly selected pair of the branching points for the individual 3D volume images of IPVLs in Figs 1 and S1, and these diameters were averaged for each maturation stage of IPVLs.

Densitometric analyses of the CBB-stained gels and immunoblots were performed using the same ImageJ 1.47n software (Analyze > Plot Profile or Analyze > Gels). For the immunoblots, it was confirmed that all analyzed signals on the same membrane were not saturated in brightness to be subjected to the analyses. Chi-square tests were used to evaluate differences in the relative signal-intensity ratios of the 2D-PAGE spots of ZP3 isoforms among distinct IPVLs, although all the ratios were doubled to be tested to follow Cochran's rule [30].

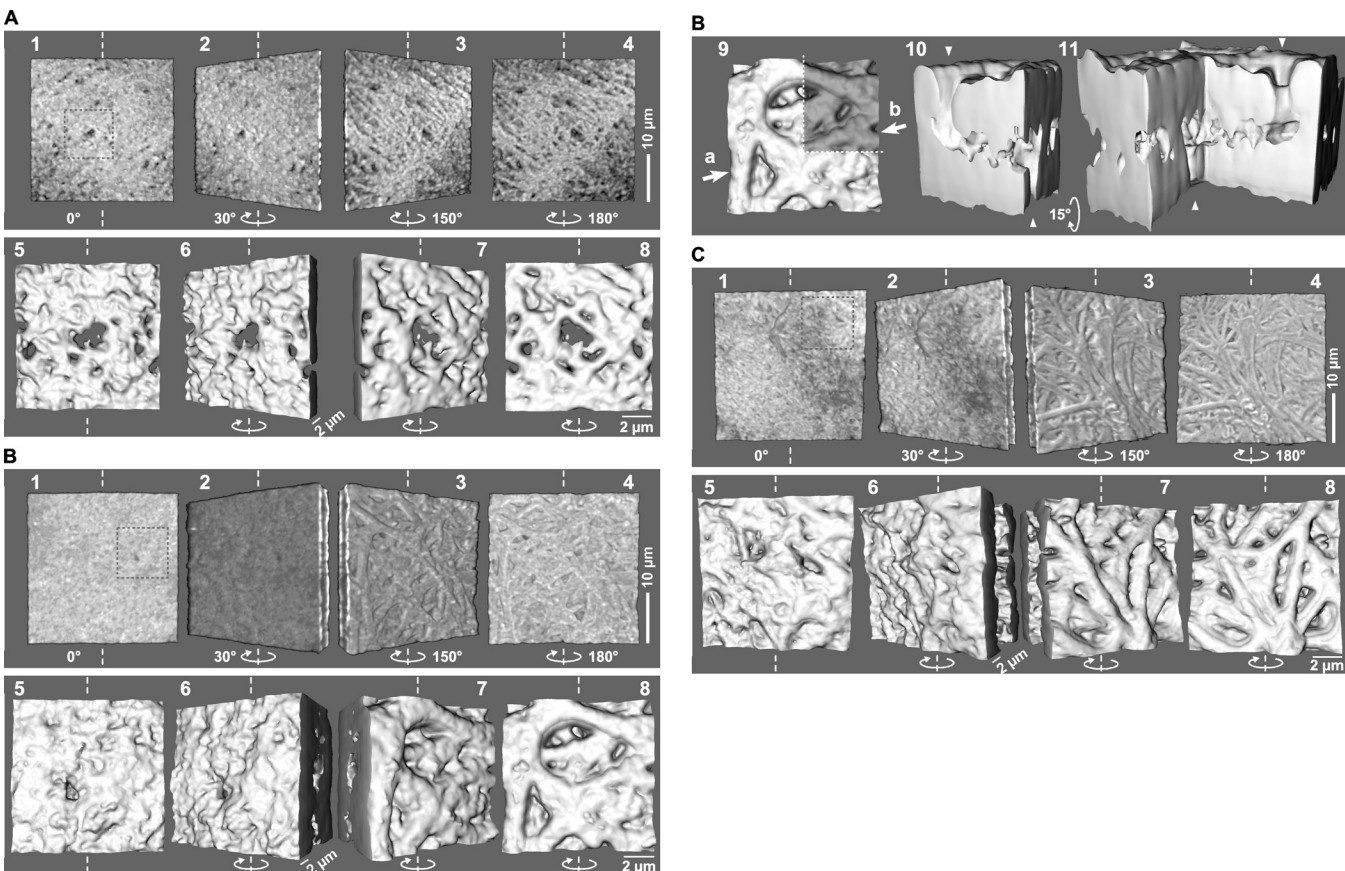

**Fig 1. 3D-surface morphologies of egg-coat matrices in IPVLs isolated from different growing stages of ovarian follicles.** IPVLs of the commercial White-Leghorn hens were isolated from the yellow follicles F3 (A), F2 (B), and F1 (C), and 3D images of them were obtained by the indirect immunofluorescence staining with the primary antibody against the repeat (avian Pro/Thr/Leu-rich) region of chicken ZP1 and the Alexa Fluor® 488-labeled secondary antibody, followed by a confocal laser scanning microscopy. Thirty-micrometer squares of the IPVLs were rendered as the 3D-volume images with rotational angles of 0˚, 30˚, 150˚, and 180˚ indicated by *white elliptic arrows* from one surface displaying smooth or homogeneous microstructures to the other surface displaying fibrous microstructures along the *vertical white dashed lines* (*panels* 1–4, respectively). Ten-micrometer squares shown by *gray dotted lines* inside the 30-μm square images (*panels* 1) were rendered as the magnified 3D-surface images with similar rotational angles (*panels* 5–8, respectively). For the F2 IPVL (B), the cross-section along the *white dotted lines* in the *panel* 9 was also rendered as 3D-surface images from the viewpoints shown by *white arrows* a and b in it with look-down angles of 15˚ indicated by *white elliptic arrows* (*panels* 10 and 11, respectively; *white arrowheads* indicate the traversing tunnel). *Thicker bars* (*panels* 4): 10 μm. *Thinner bars* (*panels* 6 and 8): 2 μm.

## Results

### Visualization of structural differences of the egg-coat matrices among chicken IPVLs at different maturities

3D images of the egg-coat matrices were obtained for IPVLs of the commercial White-Leghorn hens by the indirect immunofluorescence staining using the antibodies against chicken ZP1 followed by a confocal laser scanning microscopy to visualize changes of the egg-coat matrix in appearance (Figs 1 and S1) and internal structure (Figs 2 and S2) during the rapid growth of chicken ovarian follicles. IPVLs of the yellow follicles F3 (Figs 1A and S1A), F2 (Figs 1B and 2A and S1B–S1F and S2A–S2C), and F1 (Figs 1C and 2B and 2C and S1G and S2D) were analyzed. For all the analyzed pieces of IPVLs, several areas of 30-μm squares were rendered as the 3D-volume images (*panels* 1–4 in both Figs 1 and S1, and *panels* 1 in both Figs 2 and S2), and the areas shown by *gray dotted lines* (10-μm squares or the individually defined areas) inside these 30-μm square images were rendered as the magnified 3D-surface images (*panels* 5–8 in both Figs 1 and S1, and *panels* 2, 2–3, or 2–4 in both Figs 2 and S2). Differential-interference-contrast (DIC) microscopy images of the aforementioned 30-μm square areas in the IPVLs from F2 follicles were obtained as the controls focusing on slightly above one surfaces (*panels* 1 in both S1D and S1F Fig, corresponding to *panels* 1 in both S1C and S1E Fig, respectively) and on the other surfaces (*panels* 2 in both S1D and S1F Fig, corresponding to *panels* 4 in both S1C and S1E Fig, respectively). For the F2 IPVLs, cross sections along the *white dotted lines* in the aforementioned 10-μm square images (*panels* 9 in Fig 1B, and both S1B and S1C Fig) were also rendered as 3D-surface images (*panels* 10 and 11 in the same Figs).

The 3D images of IPVLs with rotation along the vertical axes (indicated by *white dashed lines* and *elliptic arrows* in *panels* 1–8 of both Figs 1 and S1) showed that one surfaces of the IPVLs displayed smooth or homogeneously granular microstructures (*panels* 1 and 5 with rotational angles of 0˚, and *panels* 2 and 6 with that of 30˚), whereas the other surfaces of IPVLs displayed fibrous microstructures (*panels* 3 and 7 with rotational angles of 150˚, and *panels* 4 and 8 with that of 180˚). The approximate thicknesses of IPVLs in the F3, F2, and F1 follicles were ~2, ~7, and ~7 μm, respectively, and the average diameters of the surface-exposed fibers on IPVLs in the F3, F2, and F1 follicles (mean ± SD) were 0.75 ± 0.12, 1.4 ± 0.30, and 1.5 ± 0.58 μm, respectively. Although the approximate thickness and average fiber diameters of the F2 and F1 IPVLs were similar, several holes and cavities were predominantly found in the F2 but not F1 IPVLs. In many cases, tunnels traversing the IPVLs (indicated by *white arrowheads*) were found in the F2 follicles (*panels* 10 and 11 in Figs 1B and S1B and S1C; cross sections of *panels* 9 being seen from the direction of *white arrows* a and b with look-down angles of 15˚, 0˚, and 30˚ indicated by *white elliptic arrows*, respectively). Meanwhile, in some 3D images of the above-mentioned 30-μm square areas of the F2 IPVLs, there were dark regions with weaker fluorescence on the smooth surfaces (*panels* 1 and 2 in S1C and S1E Fig). The shapes of these dark regions were approximately equal to the outlines of debris on the smooth surfaces of the F2 IPVLs as compared to the corresponding DIC images (*panels* 1 in both S1D and S1F Fig).

Interestingly, the 3D surface images of the mechanically broken sites of the F2 IPVLs showed that there were vertically oriented fibrous microstructures between their surface layers of them (*panels* 2 and 3 in Fig 2A from the viewpoints shown by *white arrows* in *panel* 1 with look-down angles of 15˚ and 30˚ indicated by *white elliptic arrows*, respectively, and *panels* 3 and 4 in S2A–S2C Fig from the viewpoints shown by *white arrows* a and b in the *panels* 2 with look-down angles of 15˚, 15˚ indicated by *white elliptic arrows* and 0˚, respectively). On the other hand, the 3D-surface images of the mechanically broken sites of the F1 IPVLs showed that there were at least three layers of spherical protrusions, including both surface layers (*panels* 2 of Figs 2B and

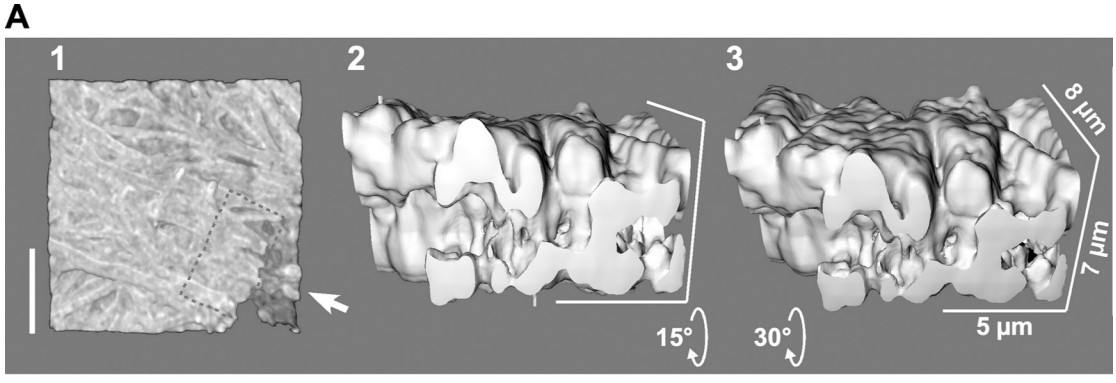

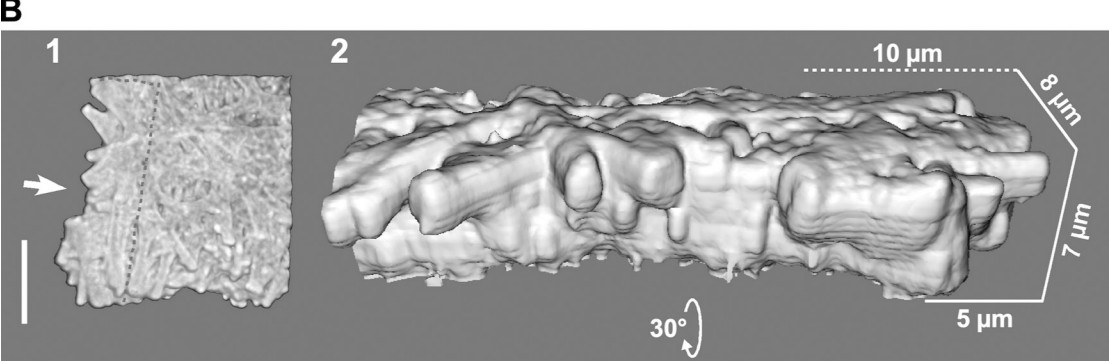

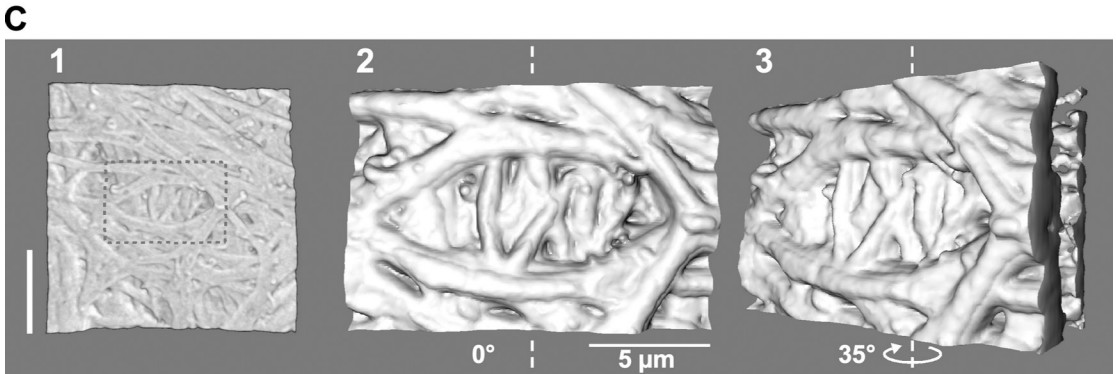

**Fig 2. Inner structures of egg-coat matrices observed in mechanically broken sites and crevice-like structure on IPVLs.** IPVLs of the commercial White-Leghorn hens were isolated from the yellow follicles F2 (A) and F1 (B and C), and 3D images of them were obtained similarly to the Fig 1, except that the primary antibody against the trefoil domain of chicken ZP1 was used for B. Approximately 30-μm squares of the IPVLs containing mechanically broken sites (A and B) and a crevice-like structure between thicker fibers (C) were rendered as the 3D-volume images on the fibrous surface (*panels* 1). The areas shown by *gray dotted lines* inside *panel* 1 were rendered as the magnified 3D-surface images from the viewpoints shown by *white arrows* in them with look-down angles of 15˚ (*panel* 2 in A) and 30˚ (*panel* 3 in A and *panel* 2 in B), and with rotational angles of 0˚ and 35˚ from the fibrous surface (indicated by *white elliptic arrows*) along the *vertical white dashed lines* (*panels* 2 and 3 in C, respectively). *Thicker bars* (*panels* 1): 10 μm. *Thinner bar* (*panel* 2 in C): 5 μm. The *3D scale bars* are also shown (*panels* 2 and 3 in A; and *panel* 2 in B).

S2D from the viewpoints shown by *white arrows* in the *panels* 1 and look-down angles of 30˚ indicated by *white elliptic arrows*). Additionally, a 3D image of F1 IPVL with rotation along the vertical axes showed that some of the lateral thinner branches of the two thicker fibers extended to the deeper region of the other thicker fiber each other to form a crevice-like structure (*panels* 2 and 3 of Fig 2C with rotational angles of 0˚ and 35˚ indicated by *white elliptic arrows*, respectively, from the fibrous surface along the *vertical white dashed lines*).

## Determination of relative proportions of ZP3 isoforms in the chicken IPVLs of different maturities

Proteins in the individual chicken IPVLs that were collected from the yellow follicles F1 to F3 of the commercial White-Leghorn hen or the long-term closed colony of White-Leghorn breed (WL-G) hen were separated by 2D-PAGE, and proteins in the 2-D gels were detected by silver staining, CBB staining, and immunoblotting using the anti-ZP3 antiserum (Figs 3A and S3A, and *lower panels* in S3B–S3E Fig). 2D-PAGE and silver staining of an individual IPVL from the F1 follicle of the commercial White-Leghorn hen confirmed that at least 9 isoforms of ZP3 (designated as ZP3 isoforms 1 to 9 from acidic to basic, respectively) were contained in the individual IPVLs of mature yellow follicles as in our previous report [25] (*panel* 1 in Fig 3A that were extracted from S3A Fig, with vertical lines indicating each spot of the ZP3 isoforms). The unidentified but minor proteins (*single* and *double asterisks* in both *panel* 1 of Figs 3A and

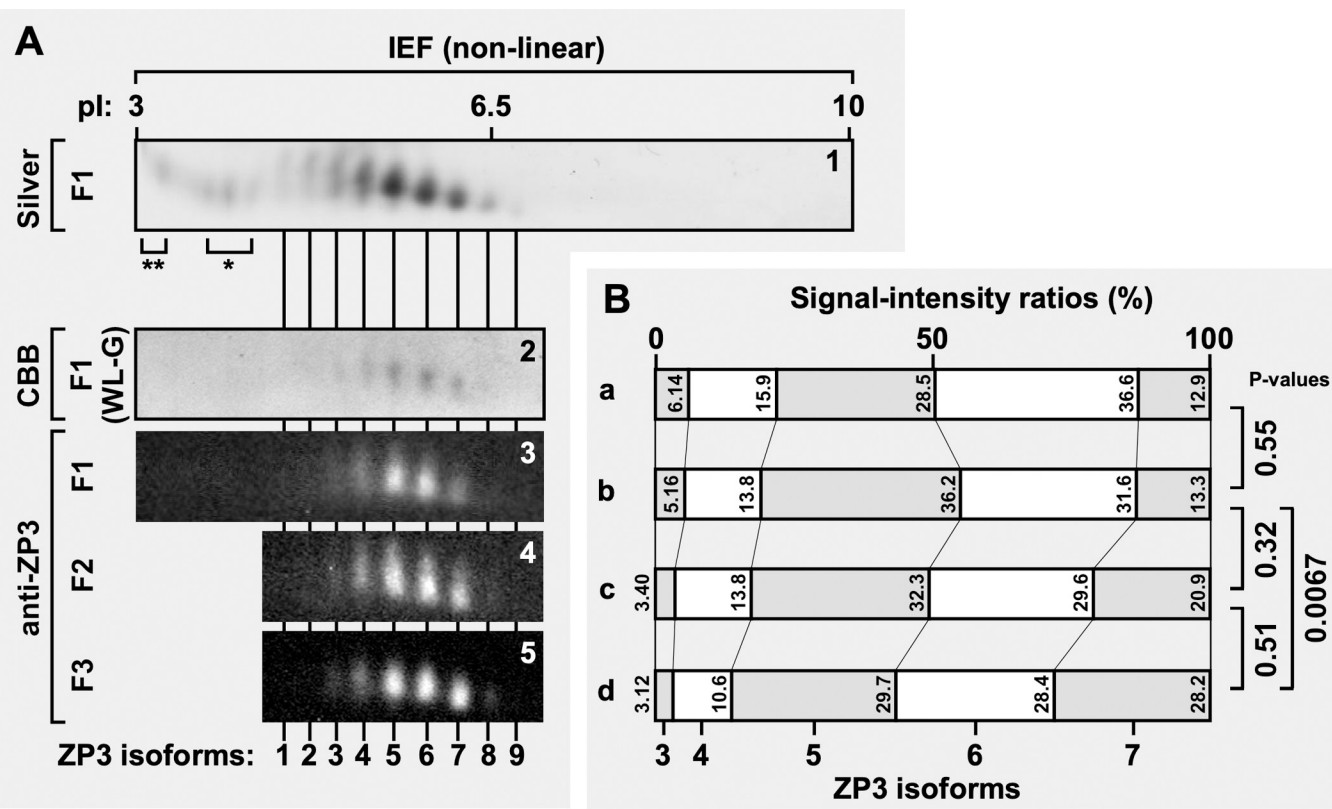

**Fig 3. Relative proportions of ZP3 isoforms in the chicken IPVLs of different maturities.** (A) Proteins in the chicken IPVLs collected from the yellow follicles F1 to F3 of the commercial White-Leghorn (*panels* 1 and 3 to 5) or the WL-G (*panel* 2) hen were separated by 2D-PAGE. All the proteins in the F1 IPVL of the commercial White-Leghorn hen were detected by silver staining (*panel* 1), although the 2-D gel image containing the ZP3 isoforms (~36 kDa) was extracted along the horizontal length of the gel ranging from pI 3 to 10. *Single* and *double asterisks* in *panel* 1 indicate unidentified proteins also detected in our previous study [25]. Proteins in the F1 to F3 IPVLs of the commercial White-Leghorn hen were also detected by immunoblotting using the anti-ZP3 antiserum (*panels* 3 to 5, respectively). The blot images are extracted similarly to *panel* 1, ranging from pI 3 to ~7 (*panel* 3) or pI ~4 to ~7 (*panels* 4 and 5). In addition, proteins in the F1 IPVL of the WL-G hen were detected by CBB staining (*panel* 2), although the 2-D gel image was extracted similarly to *panel* 3. The horizontal positions of the spots of ZP3 isoforms are indicated on the bottom of *panel* 5 with *vertical lines*. Theoretical pIs are shown on the top of *panel* 1. (B) Densitometric analyses for the ZP3 isoforms against the gel and blot images of *panels* 2 to 5 in A were performed. The densitometric values for the spots of the ZP3 isoforms were calculated as in S3B–S3E Fig, and the signal-intensity ratios of the major ZP3 isoforms 3 to 7 in each IPVL are displayed by the band graphs, respectively (*graphs* a to d corresponding to *panels* 2 to 5, respectively). The band positions for each isoform are shown at the bottom of *graph* d, and the values of signal-intensity ratio of each isoform are shown in the graphs. Proportions of the signal intensities from ZP3 isoforms 3 to 7 are compared between F1 IPVLs of the commercial White-Leghorn and WL-G hens, and between F1 and F2, F2 and F3, and F1 and F3 IPVLs of the commercial White-Leghorn hens, respectively, by chi-square test (P-values on the right side of the graphs).

S3A) that were non-immunoreactive with the anti-ZP3 (see *panel* 3 in Fig 3A) were also detected as in our previous report [25]. According to 2D-PAGE of the F1 IPVLs of the WL-G hen followed by CBB staining (*panel* 2 in Fig 3A) and that of the commercial White-Leghorn hen followed by the immunoblotting (*panel* 3 in Fig 3A), the ZP3 isoforms 3 to 7 were major components in the F1 IPVLs, whereas the ZP3 isoforms 1, 2, 8 and 9 were minor ones. Interestingly, the ZP3 isoform 8 was detected in the F3 and F2 follicles but undetectable in the F1 follicle (*panels* 3 to 5 in Figs 3A and S3C–S3E, respectively).

To determine relative proportions of the major ZP3 isoforms in the F1 to F3 IPVLs, densitometric analyses of the spot of ZP3 isoforms were performed (Figs 3B and S3B–S3E). For each ZP3-spot image (*lower panels* of S3B–S3E Fig), the vertically-averaged pixel intensities (the gray values) were plotted against the horizontal distances, and the peak areas were measured using the tangent-skim method (*upper panels* of S3B–S3E Fig with the peak areas). Band graphs for the signal-intensity ratios of the major ZP3 isoforms 3 to 7 in the F1 to F3 IPVLs (Fig 3B) showed that relative proportions of these ZP3 isoforms in the F1 IPVL of the WL-G hen (*graph* a) were nearly identical (P = 0.55, by chi-square test) to that of the commercial White-Leghorn hen (*graph* b). The graphs also show that the relative proportions of ZP3 isoforms 3 to 7 were significantly different between the F1 and F3 IPVLs (*graphs* b and d, respectively; P = 0.0067, by chi-square test), although they were nearly identical both between the F1 and F2 IPVLs (*graph* b and c; P = 0.32, by chi-square test) and between the F2 and F3 IPVLs (P = 0.51, by chi-square test), in the commercial White-Leghorn hen.

## Detection of distinctive amino-acid substitutions and post-translational modifications among ZP3 isoforms

To analyze structural differences among the ZP3 isoforms, proteins in the chicken IPVLs that were collected from the yellow follicles F1 of both the commercial White-Leghorn and the WL-G hens were separated by 2D-PAGE, and the major ZP3 isoforms 3 to 7 were isolated from the 2-D gels to be subjected to the tandem mass spectrometry (MS/MS). The MS/MS data were analyzed using the Mascot Server as mentioned in the Materials and Method section. The Mascot MS/MS-ions search against the obtained MS/MS datasets without the Error-tolerant search detected 7 to 14 of the ZP3-derived tryptic peptides from each ZP3 isoforms of both hens. Subsequently, the detected ZP3-derived peptides were subjected to the Manual error-tolerant search, excluding the peptides that were not detected at the significance levels of P < 0.05 for all the ZP3 isoforms. For each peptide, the matched sequence at significance levels of P < 0.05 with the highest Mascot ions score among the candidate sequences was determined as the most probable sequence, with the exception that their Mascot ions scores were less than that of the sequences in which artifactual modifications were detected. The determinations of the most probable sequences for each peptide were performed to identify as much common modifications as possible between the corresponding ZP3 isoforms of the commercial White-Leghorn and the WL-G hens. The detected peptides and their sequences were shown in the S1 Table. The locations of detected peptides and the determined modifications on the amino-acid sequence of chicken ZP3 were summarized in Fig 4A and 4B, respectively.

The MS/MS analyses showed that the peptides corresponding to residues 32–46, 126–142, 143–156, 257–270, 299–311, 318/320–329, and 330–347 (named the peptides 32–46, 126–142, 143–156, 257–270, 299–311, 318/320–329, and 330–347, respectively) were detected from all the ZP3 isoforms 3 to 7 of both the commercial White-Leghorn and the WL-G hens, regardless of the significance levels. On the other hand, the peptides corresponding to residues 47–69 and 77–91 (named the peptides 47–69 and 77–91, respectively) were detected from ZP3 isoforms 5, 6, and 4, 6, 7, respectively, of the commercial White-Leghorn but not the WL-G hens. In

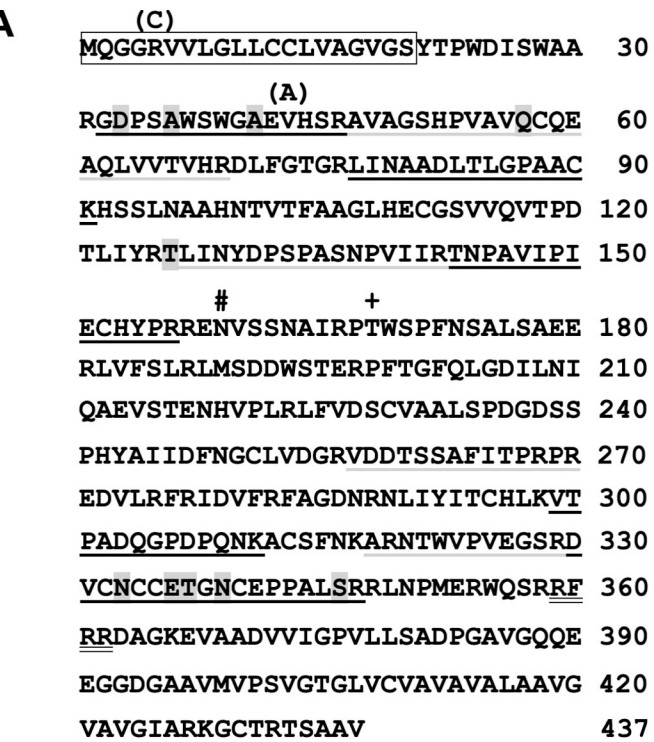

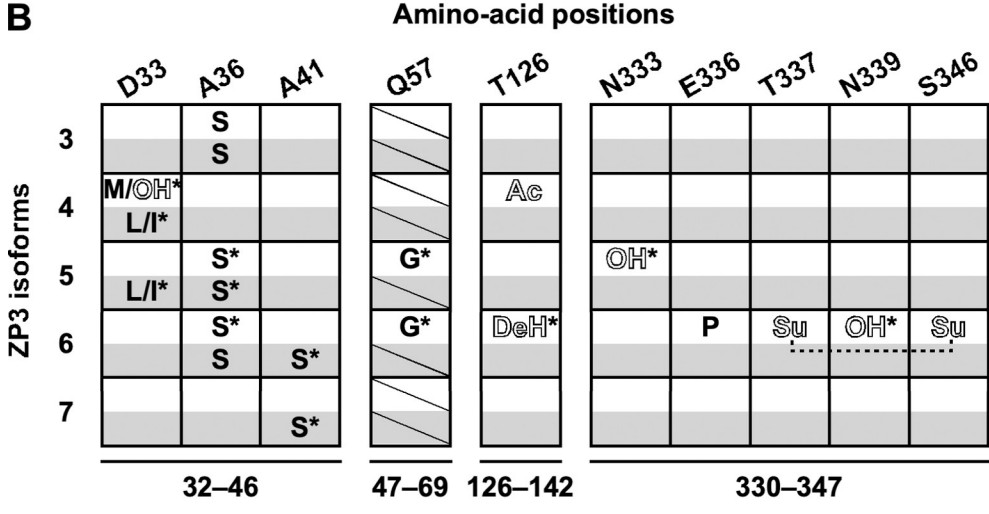

**Fig 4. Characteristic post-translational modifications and amino-acid substitutions on the ZP3 isoforms.** (A) Major ZP3 isoforms 3 to 7 were separated by 2D-PAGE of the IPVLs that were collected from the yellow follicle F1 of both the commercial White-Leghorn and WL-G hens, and analyzed by tandem mass spectrometry (MS/MS) analyses. Amino-acid sequence of chicken ZP3 is shown based on GenPept accession no. NP_989720.3, although its methionine (M) 10 is interpreted as the N-terminal methionine (M) according to GenPept accession no. BAA13760.3. The putative signal peptide and a potential consensus sequence for furin cleavage are indicated by a *rectangle* and a *double underline*, respectively, and the potential *N*-glycosylation site and the previously identified *O*-glycosylation site [31] are indicated by # and + *symbols*, respectively. The *alternate black* and *gray underlines* indicate the peptides detected by MS/MS analyses, and the amino-acid residues containing the identified modifications are *shadowed*. The C5 and A43 mutants derived from the heterozygous allele of WL-G chicken are shown in *parentheses*. (B) Amino-acid positions containing post-translational modifications (*outlined letters*) and polymorphic amino-acid substitutions (*regular letters*) that are identified at significance levels of $P < 0.05$ (*: $P < 0.01$) are summarized for ZP3 isoforms 3 to 7 of both the commercial White-Leghorn (*white boxes*) and the WL-G (*shadowed boxes*) hens. Original amino-acid positions are

shown on the top, ZP3 isoforms are indicated on the left, and peptides containing these modifications are indicated on the bottom. The peptides corresponding to residues 47–69 (the peptide 47–69) are not detected except from the ZP3 isoforms 5 and 6 of the commercial White-Leghorn hen (*boxes with a diagonal line*). *Ac*: acetylation, *DeH*: dehydration, *OH*: hydroxylation, *Su*: sulfonation.

addition to these original peptides without any modifications, the corresponding peptides containing post-translational modifications and/or amino-acid substitutions were detected in a ZP3 isoform-dependent manner as follows: the peptides 32–46 containing the modifications at one or two of the positions D33, A36, and A41 for ZP3s of both the commercial White-Leghorn and the WL-G hens, the peptides 47–69 containing the modifications at the position Q57, the peptides 126–142 containing the modifications at the position T126, and the peptides 330–347 containing the modifications at one of the positions N333, E336, N339, and the combination of the positions T337 and S346, uniquely for ZP3 of the commercial White-Leghorn hen. In the peptides 32–46, there were obvious correlations in the modifications and/or the amino-acid positions containing these modifications between ZP3 isoforms of the commercial White-Leghorn and the WL-G hens, except that the modifications at positions A41 were detected in the ZP3 isoforms 6 and 7 uniquely for the WL-G chicken. Expression of the heterozygous alleles being identified in the ZP3 gene of the WL-G chicken (GenBank accession nos. LC652422 and LC652423) were confirmed by the detection of A43V and V43A substitutions.

## Development of a mild method to produce chicken ZP1-immobilized beads to be used for the beads-based ZP1–ZP3 binding assay

Avian ZP1 molecules contain the equivalent of the Pro/Thr/Leu-rich region of mammalian ZP1s between the N-terminal ZP-N1 and the trefoil domains, although the avian Pro/Thr/Leu-rich region is generally much longer and more repetitive than mammalian one and named the repeat region in chicken [15]. Furthermore, at least in chicken ZP1, the repeat (avian Pro/Thr/Leu-rich) region contains a large number of histidine residues, most of which are arranged at approximately regular intervals of the amino-acid sequence. Therefore, we examined the binding capacity of chicken ZP1 that contains 26 to 29 histidine residues (depending on the determination of domain structure) in the repeat (avian Pro/Thr/Leu-rich) region for the nickel-affinity magnetic beads to develop a method to immobilize native chicken ZP1 on the carrier beads under mild conditions.

Previous studies have shown that avian (chicken and quail) ZP1s are expressed in the liver and secreted in the blood to be delivered to the ovary and incorporated in the egg coat matrices [20–23]. Hence, we incubated the nickel-affinity magnetic beads with the laying-hen serum containing ZP1, and after washing, the proteins being bound to the beads were eluted with the imidazole-containing buffer (Figs 5 and S4). Non-reducing SDS-PAGE of the serum, the flow-through, the wash and elution fractions followed by the silver staining or immunoblotting using the anti-ZP1 antiserum showed that ZP1 in the laying-hen serum that was detected as an ~97-kDa monomer was selectively bound to the beads and eluted with the imidazole-containing elution buffer together with some other serum proteins.

## Detection of distinct differences in the ZP1 binding affinities among ZP3 isoforms

IPVLs that were collected from the yellow follicles F1 of the commercial White-Leghorn hens were subjected to the preparative liquid-phase isoelectric focusing (LP-IEF) using carrier ampholytes for both pH ranges of ~3.5–9.5 and ~5–7 after the removal of ZPD by ultrasonic treatment [18] and fractionated individually into 10 fractions (S5A and S5F Fig, respectively).

Non-reducing SDS-PAGE of the collected fractions showed that ZP3 was concentrated at least in the fractions 5 and 6 for the pI range of ~3.5–9.5 (S5A Fig) and in the fractions 7 to 9 for the pI range of ~5–7 (S5F Fig). Each of these fractions were pooled and subjected to the subsequent re-fractionation procedure, and fractionated individually into 10 fractions (Fig 6A and 6E, respectively). Non-reducing SDS-PAGE of the fractions obtained from the re-fractionation showed that ZP3 in the pooled fractions 5 and 6 for the pI range of ~3.5–9.5 were separated into the fractions 3 to 10 (Figs 6A and S5B). 2D-PAGE images showed that the fractions 3, 7, and 10 that were obtained from the re-fractionation mainly contained the ZP3 isoforms 2 to 4, 5 to 7, and 7 to 8, respectively, and these fractions were renamed as the crude ZP3-isoform fractions a, b and c, respectively (Figs 6B and S5C). Similarly, non-reducing SDS-PAGE of the fractions obtained from the re-fractionation showed that ZP3 isoforms in the pooled fractions 7 to 9 for the pI range of ~5–7 were separated into at least the fractions 5 to 9 (Figs 6E and S5G). 2D-PAGE images showed that the fractions 6, 8, and 9 that were obtained from the re-fraction-ation mainly contained the ZP3 isoforms 5, 6 to 7, and 7, respectively, and these fractions were renamed as the crude ZP3-isoform fractions d, e, and f, respectively (Figs 6F and S5H).

Subsequently, the ZP1–ZP3 binding assays were performed as mentioned in the Materials and Methods section. The crude ZP3-isoform fractions a–c (*lanes* labeled as a–c under "Crude ZP3" in Fig 6C) and d–f (*lanes* labeled as d–f under "Crude ZP3" in Fig 6G), and the corresponding elution fractions a–c (*lanes* labeled as a–c under "Elution" in Fig 6C) and d–f (*lanes* labeled as d–f under "Elution" in Fig 6G) were subjected to non-reducing SDS-PAGE followed by silver staining (S5D and S5I Fig, respectively) and immunoblotting (Figs 6C and S5E and 6G and S5J, respectively). The densitometric analyses were performed only for the known signals of both ZP1 and ZP3 but not for the unidentified signals of both the anti-ZP1 and anti-ZP3 antisera that were detected in S5E and S5J Fig (*single*, *double*, and *triple asterisks*). The densitometric values of ZP3 signals from both the crude ZP3-isoform and the elution fractions, and that of monomeric ZP1 signals from the elution fractions, were measured for the fractions a–c (Fig 6C) and d–f (Fig 6G). The ratios of the densitometric values of ZP3 signals in the

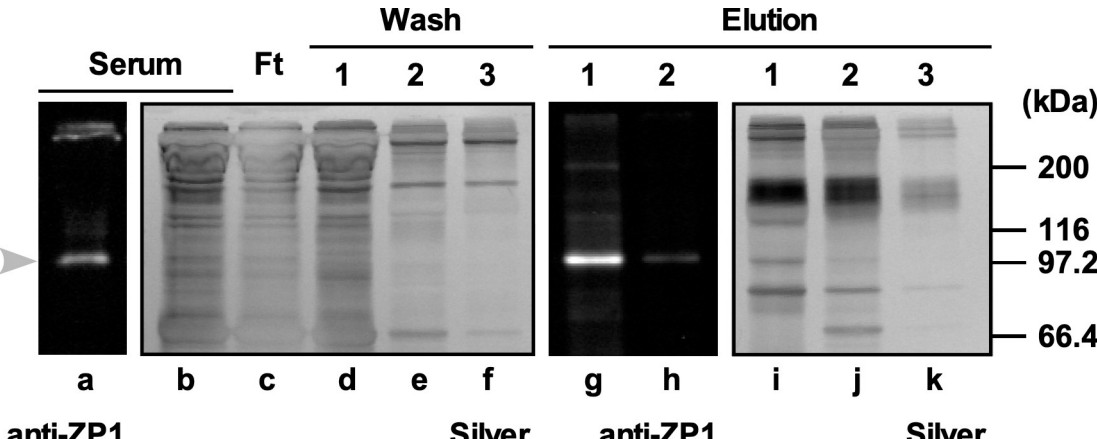

**Fig 5. Selective immobilization of chicken ZP1 on nickel-affinity magnetic beads.** To establish a method to immobilize native chicken ZP1 on the carrier beads under mild conditions, the binding capacity of chicken ZP1 containing plenty of His residues for the nickel-affinity magnetic beads was confirmed. The nickel-affinity magnetic beads were incubated with the laying-hen serum, and after washing, the proteins being bound to the beads were eluted with the imidazole-containing buffer. The results of non-reducing SDS-PAGE followed by the silver staining (*lanes* b–f and i–k) and immunoblotting using the anti-ZP1 antiserum (*lanes* a, g and h) for the serum (*lanes* a and b, labeled as "Serum"), the flow-through fraction (*lane* c, labeled as "Ft"), the wash fractions 1–3 (*lanes* d–f, labeled as "Wash") and the elution fractions 1, 2 and/or 3 (*lanes* g–k, labeled as "Elution") are shown. The migration position of the monomer of ZP1 is shown on the left side of *lane* a (*gray arrowhead*), and that of the MW markers are shown on the right side of *lane* k.

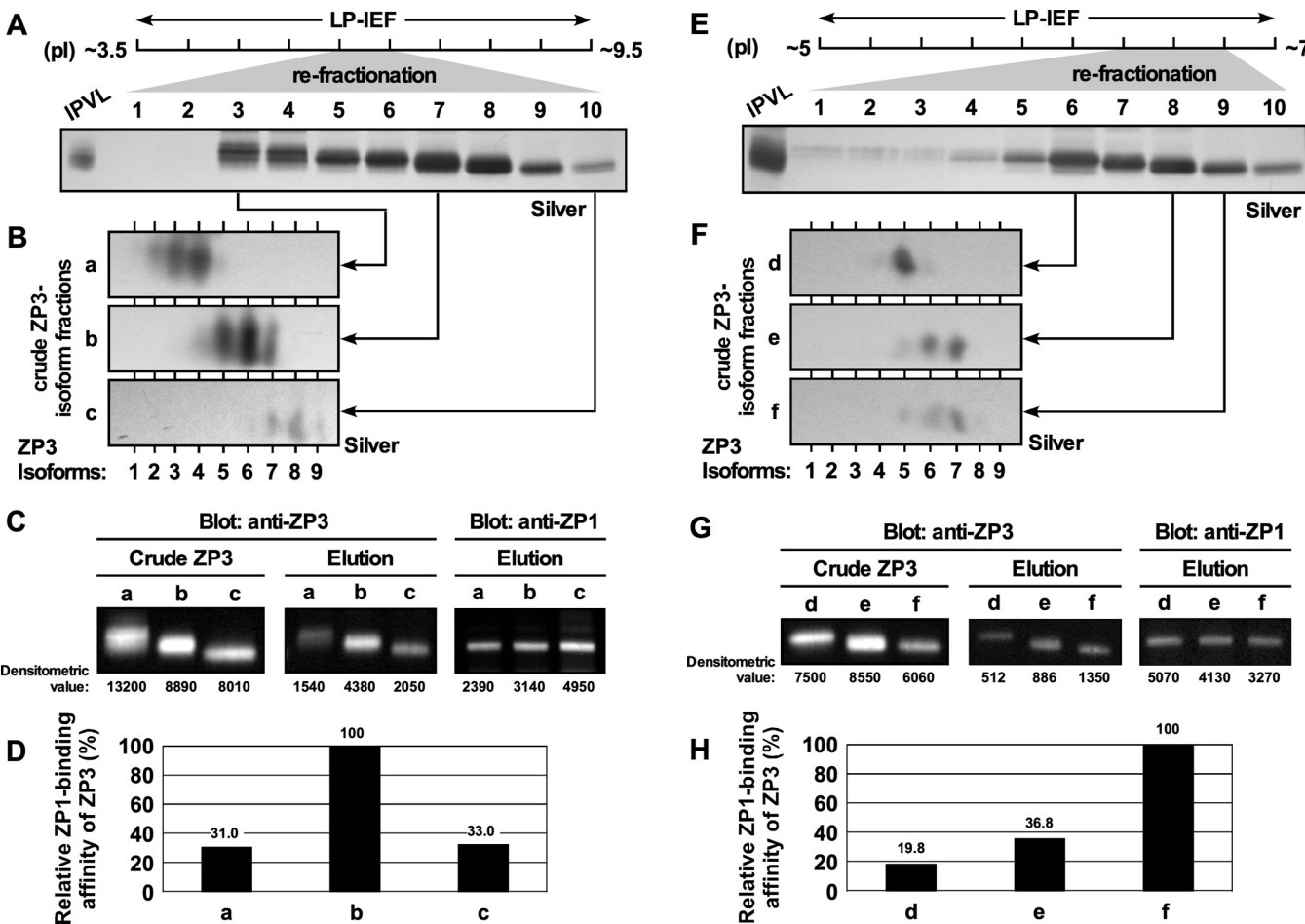

**Fig 6. Distinct differences in the ZP1-binding affinities among ZP3 isoforms.** IPVL of the commercial White-Leghorn hens that were isolated from the yellow follicle F1 were subjected to the preparative liquid-phase isoelectric focusing (LP-IEF) using the carrier ampholyte for pH ranges of ~3.5–9.5 followed by the re-fractionation procedure (A and B), and ZP1-binding affinities of ZP3 isoforms in the fractions were assayed as mentioned in the Materials and Methods section (C and D). Similarly, ZP1-binding affinities were assayed for ZP3 isoforms that were fractionated by LP-IEF of the IPVL using the carrier ampholyte for pH ranges of ~5–7 followed by the re-fractionation (E–H). (A and E) Silver-stained non-reducing SDS-PAGE gel images of each 10 fractions obtained from the re-fractionation procedures are extracted to contain ZP3 bands (~36 kDa). IPVLs were subjected to the left-side *lanes* of the fractions 1 as controls. The approximate pH ranges and fraction numbers of the re-fractionation are shown on the top. (B and F) Silver-stained 2D-PAGE gel images of the fractions 3, 7, and 10 that are seen in A (renamed as the crude ZP3-isoform fractions a, b, and c, respectively) and the fractions 6, 8, and 9 that are seen in E (renamed as the crude ZP3-isoform fractions d, e, and f, respectively) are extracted along the horizontal length of the gel ranging from pI ~4 to ~7 to contain ZP3-isoform spots (~36 kDa). The horizontal positions of the spots of ZP3 isoforms are indicated on the bottom with *vertical lines*. (C and G) Immunoblots for both the crude ZP3-isoform fractions (the left 3 *lanes* labeled as "Crude ZP3") and the elution fractions (middle 3 *lanes* labeled as "Elution") using anti-ZP3 antiserum, and that for the elution fractions (the right 3 *lanes* labeled as "Elution") using anti-ZP1 antiserum are extracted to contain ZP3 (~36 kDa) or ZP1 (~97 kDa) signals. The densitometric values for the blots are shown on the bottom. (D and H) The "Elution"/"Crude ZP3" ratios of the densitometric values of ZP3 signals are calculated and normalized for the corresponding values of ZP1 signals in the "Elution". The relative ratios are displayed by *bar graphs* as the "relative ZP1-binding affinity of ZP3 (%)". The values of relative ratios are shown on the top of the *bars*.

elution fractions against the crude ZP3-isoform fractions were calculated for all the fractions a–f, and normalized for the densitometric values of ZP1 signals in the corresponding elution fractions, respectively, as the ZP1 binding affinities of ZP3. The relative ratios of the ZP1 binding affinities of ZP3 in the fractions a–c (Fig 6D) showed that ZP3 in the mixture of ZP3 isoforms 5–7 (major components in the fraction b) bound to ZP1 with approximately 3-times higher affinity than ZP3 in the mixtures of ZP3 isoforms 2–4 and 7–8 (major components in the fractions a and c, respectively), although the non-ZP3 contaminants in the crude ZP3-isoform fractions a–c (see S5D Fig) might interact with the ZP1-beads. Whereas that in the

fractions d–f (Fig 6H) showed that ZP3 isoform 7 (a major component in the fraction f) bound to ZP1 with approximately 5- and 3-times higher affinity than ZP3 isoform 5 (a major component in the fraction d) and ZP3 in the mixture of ZP3 isoforms 6–7 (major components in the fraction e). These results showed that the ZP3 isoform 7 exhibited much higher ZP1 binding affinity among ZP3 isoforms 2–8, although there might be possible synergistic effects of these ZP3 isoforms (e.g., the ZP1 binding activity of the isoform 7 might be inhibited by the isoform 6 and be enhanced by the isoform 5).

## Discussion

### Asymmetrical formation of the egg-coat matrix in the quickly growing chicken IPVL

The vertebrate egg coat, including mammalian zona pellucida (ZP), avian inner-perivitelline layer (IPVL), and so on, is composed of a fibrous network that is formed by the ZP glycoproteins [32, 33]. In the previous studies, the high-resolution scanning electron microscopy showed distinct differences in ultrastructures between the immature and mature egg-coat matrices of both human and bovine ZP, and between the inner and outer surfaces of human ZP [34, 35]. In the present study, we analyzed the 3D surface structures of chicken IPVL in the three distinct maturities under wet conditions by using immunofluorescent staining followed by confocal laser scanning microscopy. Chicken IPVL is composed of a more robust network and can be more easily isolated from the ovarian follicles at macroscopically distinguishable stages of development than mammalian ZP in general.

In the immature IPVLs that are isolated from the yellow follicles F3, there are distinct differences in ultrastructures between the inner and outer surfaces, i.e., one surface has a smooth or homogeneously granular appearance (*panels* 1, 2, 5, and 6 in both Figs 1A and S1A), while the other surface is composed of fibrous structures (*panels* 3, 4, 7, and 8 in both Figs 1A and S1A). Although the mechanisms underlying such asymmetrical formation of the egg-coat matrix is not clear, it is possible that the assembly of the egg-coat matrix is initiated at the one surface of IPVL or ZP adjacent to the ZP-glycoprotein secreting cell(s) in the ovary (i.e., the granulosa cells in birds and mainly the oocyte in mammals) and proceeds to the other surface being accompanied with the formation of fibrous structures. In addition, considering that the IPVL of more immature F4 follicle is too fragile to be isolated as solids although there are certain amounts of ZP1, ZP3, and ZPD [24], the assembly of insoluble egg-coat matrix from the soluble or loose complexes of these ZP glycoproteins may occur during the growth of the yellow follicle F4 to F3.

3D images of the surfaces of F2 IPVLs (Figs 1B and S1B) showed that the approximate thickness of egg-coat matrices and the average diameters of surface-exposed fibers are increased up to approximately 3 and 2 times, respectively, during the rapid growth of ovarian follicles from the F3 to F2. The differences in the ultrastructure between both surfaces of the F2 IPVLs are roughly similar to that of the F3 IPVLs, implying that the quick development of the egg-coat matrix may be mainly associated with the extension and thickening of individual fibrous structures rather than the de novo formation of additional fibers. In the F2 IPVLs, the egg-coat matrices show comparably looser structures than the F1 IPVLs, and interestingly, some cavities and tunnels are traversing the matrices (*panels* 9–11 in Figs 1B and S1B and S1C). Some of these cavities and tunnels in the F2 IPVLs may be occupied by the avian equivalent of mammalian transzonal projections (TZPs) that are involved in the intercellular communications between the growing oocyte and the granulosa cells (cumulus cells) in the ovarian follicles [36, 37]. Additionally, the shape of dark regions with weaker fluorescence signals on the 3D volume images of the smooth surfaces of F2 IPVLs (*panels* 1 and 2 in both S1C and S1E

Fig) correspond to the shape of the probable debris that are observed only on the same surface by differential-interference-contrast (DIC) microscopy (*panels* 1 in both S1D and S1F Fig). These results imply that the smooth surface of IPVL may be adjacent to the granulosa cell layer in the ovarian follicles, although further analyses to determine the orientation of IPVL on the oocyte are required.

The morphological appearances of the egg-coat matrices in the F1 IPVLs are nearly identical to that in the F2 IPVLs (Figs 1C and S1G). However, the deep regions of the F1 IPVLs are much less stained than that of F2 IPVLs (compare *panels* 6 and 7 of Figs 1B and S1B, S1C and S1E to that of Figs 1C and S1G), although all the IPVLs were pretreated with the 0.1% Triton X-100 as mentioned in the Materials and Methods section. Furthermore, the aforementioned cavities and tunnels traversing the egg-coat matrices are found in the F2 IPVLs but not in the F1 IPVLs at all. These results together with the observation that the F1 IPVLs are harder and easier to be detached from the ovarian follicles than the F2 IPVLs, suggest that the loose F2 IPVLs mature quickly into the F1 IPVLs in which the fibrous structures form tight associations with each other showing much lower permeability against macromolecules including antibodies than in the F2 IPVLs, and the avian equivalent of mammalian TZPs that may traverse the F2 IPVL are closed before the ovulation in the F1 IPVLs, as partly mentioned in [14].

Interestingly, 3D images of the internal structures of the egg-coat matrices were occasionally obtained from the mechanically broken sites on the IPVLs that are isolated from the F2 (Figs 2A and S2A–S2C) and F1 (Figs 2B and S2D) follicles. These images suggest that the fibrous structures in F2 IPVLs are organized vertically from the smooth surface and exposed on the other surface to be arranged horizontally, while the fibrous structures in the F1 IPVLs between both surfaces appear to be more horizontal than that in the F2 IPVLs. In addition, some ladder-like structures, in which some of the thinner fibers branch vertically from each of the two thicker fibers and extend into deeper regions of the other thicker fiber, were observed mainly on the fibrous surfaces of F1 IPVLs (Fig 2C and *panels* 3, 4, 7, and 8 in S1G Fig). These observations suggest that the aforementioned transition of the internal structures of egg-coat matrices from the F2 to F1 IPVLs may be due to the lateral extension of the F2 IPVLs with the continuous thickening of the fibers being accompanied by the mechanical tension during the rapid growth of the ovarian follicles from F2 to F1.

## Significant changes in the compositions of ZP3 isoforms associated with the maturation of chicken IPVLs

Our previous studies have suggested that the F1 IPVL of the commercial White-Leghorn hen contains at least 9 isoforms of ZP3 that exhibit distinct isoelectric points (designated as ZP3 isoforms 1 to 9 from acidic to basic, respectively; see *panel* 1 in Figs 3A and S3A). The charge microheterogeneities of proteins, including chicken ZP3, are probably due to post-translational modifications [25–27]. In the present study, our results confirm that the ZP3 isoforms 3 to 7 are major components of ZP3 in the F1 IPVLs of both the commercial White-Leghorn hens (*panel* 3 in Figs 3A and S3C) and the long-term closed colony of White-Leghorn breed (WL-G) hens (*panel* 2 in Fig 3A). Notably, the relative proportions of the ZP3 isoforms 3 to 7 in these F1 IPVLs are nearly identical between the commercial White-Leghorn and the WL-G hens (*graphs* a and b in Fig 3B). These results demonstrate that ZP3 isoforms (at least the isoforms 3 to 7) are expressed altogether in individual follicles in White-Leghorn line-specific manner of regulation, but not expressed as a result of potential genetic heterogeneities in laying hens [38, 39]. Furthermore, our results confirm that both the F2 and F3 IPVLs may also contain the ZP3 isoforms 3 to 8 being identical to

that in the F1 IPVL, since the pI profiles of these ZP3 isoforms in IEF separation are highly consistent across F1 to F3 IPVLs (Figs 3A and S3B–S3E). Interestingly, we revealed in the present study that the relative proportions of ZP3 isoforms 3 to 7 are gradually shifted during the follicular development to show significant differences between the F3 and F1 follicles, and that the ZP3 isoform 8 in the F3 and F2 follicles becomes undetectable in the F1 follicle (*panels* 3–5 in Fig 3A corresponding to S3C–S3E Fig, respectively). Considering that chicken ZP3 is expressed predominantly in the granulosa cells of ovarian follicles, these data suggest that the expression levels of individual ZP3 isoforms in the granulosa cells, and/or the populations of granulosa cells that express each ZP3 isoform, are altered during the rapid growth of the yellow follicles.

### Isoform-dependent ZP3 modifications that are partially conserved between the two White-Leghorn lines

Our previous study suggests that the ZP3 isoforms 4 to 7 in the F1 IPVL carry carbohydrate chains that show distinctively different lectin-binding specificities [25]. In this study, we further examined the differences in modifications among ZP3 isoforms in the F1 IPVL by MS/MS analyses. Surprisingly, our data suggest that there are some modifications, including not only post-translational modifications but also amino-acid substitutions in ZP3 isoforms of at least the F1 IPVL, and that these modifications occur in an ZP3 isoform-dependent manner (Fig 4 and S1 Table). In the MS/MS analyses, all the tryptic peptides containing these modifications are detected together with the corresponding unmodified ones except for the peptide 47–69 of the isoform 6, suggesting that these modifications except for the Q57G substitution do not significantly affect the isoelectric points of ZP3 isoforms. Considering that all the detected amino-acid substitutions except for the allelic V43A and A43V mutations are not encoded by the ZP3 gene of WL-G chicken (GenBank accession nos. LC652422 and LC652423), the results suggest that these amino-acid substitutions occur through posttranscriptional mechanisms including the RNA editing [40, 41] and/or the mistranslation [42, 43]. Interestingly, in both the commercial White-Leghorn and the WL-G hens, the A36S substitutions are detected in ZP3 isoforms 3, 5 and 6, while any other modifications at the A36 positions are not detected in ZP3 isoforms 4 and 7 (Fig 4B). In addition, in both the commercial White-Leghorn and the WL-G hens, a post-translational modification and/or amino-acid substitutions are detected at the D33 position in ZP3 isoform 4 (Fig 4B). These results suggest that the patterns of ZP3 isoform-dependent modifications (including post-translational modifications and amino-acid substitutions) are partially conserved, at least in the White-Leghorn breed of chicken. Thus, the conserved modifications may play fundamental roles in chicken IPVLs. In contrast, the chicken line-specific modifications may be involved in the characteristics of IPVLs in each chicken line. However, little is known about the mechanisms regulating the physiological and structural roles of ZP3 in the egg-coat matrices. Finally, the above-mentioned peptides 47–69 are not detected in the ZP3 isoforms 3, 4, and 7 of the commercial White-Leghorn hen and all the ZP3 isoforms 3 to 7 of the WL-G hen, but detected in the ZP3 isoforms 5 and 6 of the commercial White-Leghorn hen with the Q57G substitutions (Fig 4B and S1 Table), although the peptide 47–69 without any modifications is detected only in the isoform 5 (S1 Table). These results suggest that the peptides 47–69 in the isoforms 3, 4, and 7 of the commercial White-Leghorn hen and all the isoforms 3 to 7 of the WL-G hen may constitutively contain multiple or complex modifications that are difficult to be detected by the simple MS/MS analyses.

### Distinct differences in the ZP1 binding affinities among ZP3 isoforms that might be involved in the structural maturation processes of the egg-coat matrix

In our previous studies, it has been shown that ZP1 and ZP3, i.e., the major two components constituting the egg-coat matrices of the mature chicken IPVL, spontaneously assemble into insoluble fibrous complexes in several *in vitro* experimental systems [22, 23, 28]. Hence, it is suggested that the spontaneous interactions between the ZP1 being expressed in the liver to be secreted into the blood and the ZP3 being expressed in the ovarian granulosa cells to be secreted into their extracellular spaces play critical roles in the egg-coat formation mechanisms in chicken, although the involvement of ZP3 isoforms has remained unclear. In this study, ZP3 isoforms were separated from the F1 IPVL of commercial White-Leghorn hens by the preparative liquid-phase isoelectric focusing (LP-IEF) and were subjected to the more quantitative ZP1 binding assay to be investigated whether there are any differences in the ZP1 binding affinities among them. Fortunately, chicken ZP1 exhibits high binding affinity to the nickel-affinity beads (Figs 5 and S4) probably because the chicken ZP1 molecule contains approximately 30 of histidine residues in the long and potentially flexible repeat (Pro/Thr/Leu-rich) region [15, 28]. We utilized this feature of chicken ZP1 molecules to immobilize them that were isolated from the F1 IPVL by the gel-filtration chromatography onto the carrier beads to be used for the ZP3 binding assay. The results show that the ZP3 isoform 7 by itself exhibits much higher ZP1 binding affinity among ZP3 isoforms 3 to 8 (Figs 6 and S5), although it seems that there are possible synergistic mechanisms of these ZP3 isoforms in the complex formation with ZP1, and we need to confirm the results by using pure ZP3 isoform and ZP1 preparations. Accordingly, these results suggest that the individual ZP3 isoforms probably exhibit different behaviors in the ZP1–ZP3 complex formation at least in the F1 IPVL. In addition, these results are compatible with our observations that mature chicken IPVLs seem to contain an excess of ZP3 [18, 23, 24], and certain amount of ZP3 in chicken mature IPVLs may not form heterocomplexes with ZP1 but form homopolymers.

Taken together, our results suggest that both the microstructures of egg-coat matrices (Figs 1 and 2 and S1 and S2) and the compositions of ZP3 isoforms (Figs 3 and S3) in chicken IPVL are differentially altered through the rapid growth of yellow follicles in laying hens. On the other hand, our results also suggest that individual ZP3 isoforms in the matured chicken IPVL contain different modifications in the isoform-dependent manner (Fig 4 and S1 Table) and exhibit characteristic features in the ZP1–ZP3 complex formation (Figs 6 and S5), although it remains unknown whether the modifications and the ZP1 binding affinities among the ZP3 isoforms are unchanged through the follicular development. Based on these results, we expect that the possible differences of ZP3 isoforms in the complex formation with ZP1 may be due to the microheterogeneity of ZP3 isoforms in the modifications, and alterations of individual ZP3-isoform expressions may be associated with the structural maturation processes of the egg-coat matrices during the rapid growth of yellow follicles. For example, relative proportions of the ZP3 isoform 7 among the isoforms 3 to 7 in the IPVL markedly decreased from approximately 30% to 15% through the middle (F3) to late (F1) stages of the yellow follicle development (Fig 3B). The results imply that the ZP3 isoform 7 exhibiting the highest ZP1 binding affinity among the ZP3 isoforms 3 to 7 may contribute mainly to form scaffolds of the egg-coat matrices in the F3 IPVL with probably higher density and elastic stiffness (Fig 1A and S1A). Subsequently, the other ZP3 isoforms exhibiting lower ZP1 binding affinities than the ZP3 isoform 7 may assemble on to the scaffolds to form the thicker matrices in the quickly extending F2 and F1 IPVLs with probably lower density and higher flexibility (Figs 1B and 1C and 2 and S1B–S1G and S2). The key findings of our study are summarized in Fig 7.

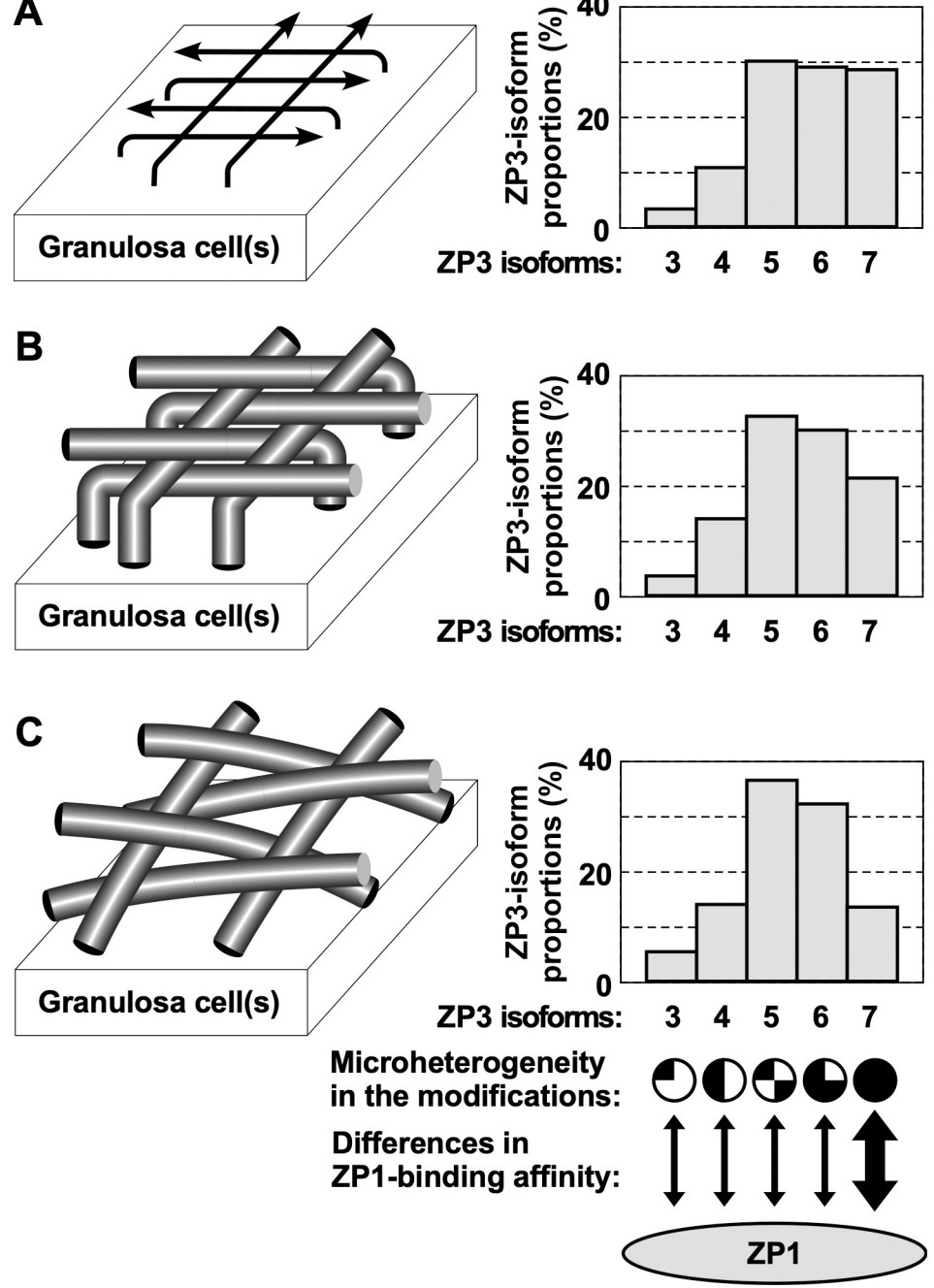

**Fig 7. Graphical summaries of key findings in this study.** (A) The *left scheme* shows that in the IPVL of yellow follicle F3, the assembly of the egg-coat matrices may be initiated at the one surface adjacent to the ZP3 secreting granulosa cell(s) and proceed to the other surface being accompanied with the formation of fibrous structures as shown by *arrows*. (B) The *left scheme* shows that the quick development of the egg-coat matrices in the IPVL of yellow follicle F2 may be mainly associated with the extension and thickening of individual fibrous structures associated with the rapid growth of the ovarian follicle. The fibrous structures in the F2 IPVLs may be organized vertically from the surface adjacent to the granulosa cell(s) and exposed on the other surface to be arranged horizontally. (C) The *left scheme* shows that in the yellow follicle F1 just before ovulation, internal fibrous structures of the egg-coat matrices appear to be more horizontal than that in the yellow follicle F2 probably because of the lateral extension of IPVLs with the continuous thickening of the fibrous structures being accompanied by the mechanical tension during the rapid growth of the ovarian follicles from F2 to F1. Interestingly, the proportions of ZP3 isoforms 3 to 7 are differentially altered through the rapid growth of yellow follicles from F3 to F1 (*bar graphs* A to C, respectively), and these ZP3 isoforms contain the isoform-dependent modifications at least in the F1 IPVL, respectively (conceptually represented

on the bottom of *bar graph* C). Furthermore, it is suggested that the ZP3 isoform 7 exhibits much higher ZP1 binding affinity among ZP3 isoforms 3 to 7, at least in the F1 IPVL (double-headed arrows on the bottom of *bar graph* C).

Overall, our results provide new insights into the functional consequences of microheterogeneity of chicken ZP3 in the egg-coat matrix formation, and it is possible that there are similar associations between the microheterogeneity of ZP glycoproteins and the egg-coat matrix formation mechanisms in all vertebrates. Nevertheless, many questions in chicken ZP3 remain to be addressed, for example, how their isoforms are produced in and secreted from the ovarian granulosa cells, although chicken ZP3 is secreted from the expressing cells forming noncovalent dimers [31], and whether their sperm binding activity [31] vary among the isoforms. Further investigations will reveal novel regulatory mechanisms of the formation of vertebrate egg coat and the sperm–egg interactions that might be associated with the microheterogeneity of ZP-glycoproteins.

## Supporting information

**S1 Fig. 3D-surface morphologies of egg-coat matrices in IPVLs isolated from different growing stages of ovarian follicles.** Other images for 3D-surface morphologies of egg-coat matrices in the F3 (A), F2 (B, C and E), and F1 (G) IPVLs are shown in the same format as Fig 1, except that the look-down angles of panels 10 and 11 in both B and C are 0˚ and 30˚ (indicated by white elliptic arrows), respectively, and that panels 9–11 that were shown in Fig 1B are not contained in E. In addition, Differential-interference-contrast (DIC) microscopy images of the 30-μm square areas for the F2 IPVLs (D and F) were obtained as the controls in focus on near the smooth surfaces (panels 1) and on the fibrous surfaces (panels 2), respectively. Bar: 10 μm.
(TIF)

**S2 Fig. Inner structures of egg-coat matrices observed in mechanically broken sites and crevice-like structures on IPVLs.** Other images for inner structures of egg-coat matrices in the F2 (A–C) and F1 (D) IPVLs are shown in the similar format as Fig 2, except that the areas shown by gray dotted lines inside the 30-μm square images for egg-coat matrices in the F2 IPVLs (panels 1 in A–C) were rendered as the magnified 3D-surface images from the viewpoints shown by white arrows a and b in panels 2 with look-down angles of 15˚ indicated by white elliptic arrows (panels 3 and 4 in both A and B) and 0˚ (panels 3 and 4 in C).
(TIF)

**S3 Fig. Relative proportions of ZP3 isoforms in the chicken IPVLs of different maturities.** (A) The original gel image of the panel 1 in Fig 3A. For comparison, proteins in the F1 IPVL of the commercial White-Leghorn hen were separated by non-reducing SDS-PAGE without IEF (right side lane). Migration positions of the disulfide-linked dimer and the monomer of ZP1 (upper and lower gray arrowheads, respectively) are shown on the left side of the gel image, and that of ZP3 (black arrowhead) and the MW markers are shown on the right side of it. Single and double asterisks are unidentified proteins also detected in our previous study [25]. (B–E) Results of densitometric analyses (upper panels) against ZP3 spot images (lower panels being identical to panels 2–5 in Fig 3A, respectively, although all are ranging from pI ~4–7) are shown. The vertically averaged pixel intensities (the gray values) were plotted against the horizontal distances, and the peak areas were measured using the tangent-skim method, although the values are reciprocal only in B. The measured peak areas are shown in the plots, and the positions of the corresponding ZP3 isoforms are shown on the bottom.
(TIF)

**S4 Fig. Selective immobilization of chicken ZP1 on nickel-affinity magnetic beads.** The original gel and immunoblot images of Fig 5, although the immunoblot images of the flow-through fraction, all the wash fractions, and elution fraction 3 are shown.
(TIF)

**S5 Fig. Distinct differences in the ZP1-binding affinities among ZP3 isoforms.** (A and F) Non-reducing SDS-PAGE gel images of each 10 fractions collected from the first step of the LP-IEFs before re-fractionation procedures using carrier ampholytes for pH ranges of ~3.5–9.5 and ~5–7, respectively, followed by silver staining. The pH ranges and fraction numbers are shown on the top of the images, and IPVLs were subjected to the left-side lanes of fractions 1 as controls. Migration positions of the disulfide-linked dimer and the monomer of ZP1 (upper and lower gray arrowheads, respectively) are shown on the right side, and that of ZP3 (black arrowhead) and the MW markers are shown on the left side of the gel images. The fractions 5 and 6 seen in A and the fractions 7 to 9 that are seen in F were pooled to be subjected to the subsequent re-fractionation procedure, respectively. (B and G) The original gel images of Fig 6A and 6E. Migration positions of ZP1 dimer and monomer, ZP3, and the MW markers are shown similarly to A. (C and H) The silver-stained 2-D gel images of all the ZP3 isoform-containing fractions 3–10 and 5–9 that are seen in Fig 6A and 6E, respectively, are extracted similarly to the Fig 6B and 6F. The extracted 2-D gel images of IPVLs are shown on the top as controls, and the horizontal positions of the spots of ZP3 isoforms are indicated on the bottom with vertical lines. (D and I) Silver-stained non-reducing SDS-PAGE gel images of both the crude ZP3-isoform fractions (labeled as "Crude ZP3") and the elution fractions (labeled as "Elution") for the fractions a–c and d–f in Fig 6B and 6F, respectively. IPVLs, the ZP1 solution and the His6-tagged thioredoxin (for blocking agent) that were used to prepare the ZP1-immobilized beads were also subjected. Migration positions of ZP1 dimer and monomer, ZP3, and the MW markers are shown similarly to A. (E and J) The original blot images of Fig 6C and 6G, although including the result of immunoblotting for the crude ZP3-isoform fraction c using anti-ZP1 antiserum. Migration positions of ZP1 dimer and monomer, ZP3, and the MW markers are shown similarly to A. The single, double, and triple asterisks indicate unidentified signals.
(TIF)

**S1 Table. Results of MS/MS analyses for ZP3 isoforms 3 to 7 isolated from IPVLs.** The peptide sequences identified with significance levels of P<0.05 are shown by *black letters*, whereas the sequences of other possible peptides are indicated by *gray letters*. Peptides containing modifications with significance levels of P<0.05 (*: P<0.01) are *shadowed*.
(XLSX)

**S1 Raw images.**
(PDF)

## Acknowledgments

We would like to thank Dr. Luca Jovine of Karolinska Institutet for helpful discussions and suggestions regarding the manuscript.

## Author Contributions

**Conceptualization:** Hiroki Okumura.

**Data curation:** Hiroki Okumura, Shunsuke Nishio.

**Formal analysis:** Hiroki Okumura.

**Funding acquisition:** Hiroki Okumura.

**Investigation:** Hiroki Okumura, Ayaka Mizuno, Eri Iwamoto, Rio Sakuma, Shunsuke Nishio.

**Methodology:** Hiroki Okumura.

**Project administration:** Hiroki Okumura.

**Resources:** Ken-ichi Nishijima.

**Supervision:** Hiroki Okumura.

**Writing – original draft:** Hiroki Okumura.

**Writing – review & editing:** Hiroki Okumura, Tsukasa Matsuda, Minoru Ujita.

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
