## [Editor Report · Decision Letter 0]

20 Jun 2022

PONE-D-22-07920

New Insights into the Role of Microheterogeneity of ZP3 During Structural Maturation of the Avian Equivalent of Mammalian Zona Pellucida

PLOS ONE

Dear Dr. Okumura

Thank you for submitting your manuscript to PLOS ONE. After careful consideration, we feel that it has merit but does not fully meet PLOS ONE’s publication criteria as it currently stands. Therefore, we invite you to submit a revised version of the manuscript that addresses the points raised during the review process.

ACADEMIC EDITOR: 

Please respond to the comments as mentioned below, and make all the necessary formatting in the revised version of this manuscript.

We look forward to receiving your revised manuscript.

Kind regards,

Birendra Mishra, DVM, PhD

Academic Editor

PLOS ONE

"This work is supported in part by Grants-in-Aid for Scientific Research (C) [Grant Number 25450520] from the Ministry of Education, Culture, Sports, Science, and Technology of Japan, and by Grants for Encouragement of Scientific Research from the Research Institute of Meijo University.  The funders had no role in study design, data collection and analysis, decision to publish, or preparation of the manuscript."

**Additional Editor comments:**

The manuscript by Okumura et al; on “New Insights into the Role of Microheterogeneity of ZP3 During Structural Maturation of the Avian Equivalent of Mammalian Zona Pellucida” is interesting and have used several techniques to prove their hypothesis. However, several important informations are missing. Authors needs to clarify the following comments.

Introduction section: It is better to include enough background information, rationale, hypothesis, and objectives of this study for better readability. Line 100-111 explains the results. Authors may consider modifying these statements.

Why is this study important, and how will it help the scientific community in the field of poultry science or biology?

Materials and Methods:

Line 118-119: Although ovary and blood were collected from a local chicken meat processor, the sampling procedure is unclear and needs a clear explanation. The number of hens, sampling, and handling procedures are missing. As this is egg-type chicken, why were samples collected from meat processors?

Line 131-133: Approximately 30 ovarian follicles for each growth stage of the yellow follicles were collected from 3 kg of 132 the ovaries of the commercial White-Leghorn hens. This statement is confusing.

Figures 1 A, B, and C look blurry, and difficult to understand. Authors may indicate with arrows or other simpler ways to understand for the readers. Same comments for Figure 2.

Figure legends (Fig 1-7) are missing. The authors have only provided the Supplementary figure legends. It’s very hard to evaluate the results without figure legends.

---

## [Author Response · Author response to Decision Letter 0]

30 Jun 2022

Thank you for reviewing our manuscript entitled “New Insights into the Role of Microheterogeneity of ZP3 During Structural Maturation of the Avian Equivalent of Mammalian Zona Pellucida”. According to reviewers’ comments, we revised our manuscript, and made answers as follows. 

Answers to the questions:

Introduction section:

>It is better to include enough background information, rationale, hypothesis, and objectives of this study for better readability. Line 100-111 explains the results. Authors may consider modifying these statements.

>Why is this study important, and how will it help the scientific community in the field of poultry science or biology?

We added sentences “Therefore, understanding the mechanisms of egg-coat formation could provide the basis for developing new approaches to control human and animal fertility including new therapies for infertility that might be related to the egg-coat abnormality.” in lines 63–65, “…. we aimed to investigate the process of egg-coat matrix formation from ZP glycoproteins using the chicken model. For this purpose, we initially collected chicken IPVLs….” in lines 102–103, and “Our results will be useful for understanding the egg-coat formation mechanisms.” in lines 118–119 of both the Revised Manuscript with Track Changes and the Manuscript.

We left line 100–111 of the initial manuscript in the revised one because this part may be useful to understand the complicated results of this study.

Materials and Methods:

>Line 118-119: Although ovary and blood were collected from a local chicken meat processor, the sampling procedure is unclear and needs a clear explanation. The number of hens, sampling, and handling procedures are missing. As this is egg-type chicken, why were samples collected from meat processors?

We replaced the word “meat processor” to more suitable one “slaughter plant” in line 127, and added sentences “In this plant, large number of White-Leghorn hens (~15,000 hens/day) are killed by bleeding from carotid artery using a mechanical knife in an automated line system and slaughtered by hand followed by a brief scalding in ~63°C of hot water for 80 seconds to be remove feathers. The blood and ovary were collected immediately and transported to our laboratory in an ice box.” in lines 127–131 of the Revised Manuscript with Track Changes and lines 127–130 of the Manuscript.

>Line 131-133: Approximately 30 ovarian follicles for each growth stage of the yellow follicles were collected from 3 kg of 132 the ovaries of the commercial White-Leghorn hens. This statement is confusing.

We replaced the sentence “Approximately 30 ovarian follicles for each growth stage of the yellow follicles were collected from 3 kg of 132 the ovaries of the commercial White-Leghorn hens,” to the one “Respective approximately 30 of yellow follicles at each growth stage (F1–F3) were collected from 3 kg of the ovaries of the commercial White-Leghorn hens,” in lines 143–144 of both the Revised Manuscript with Track Changes and the Manuscript.

>Figures 1 A, B, and C look blurry, and difficult to understand. Authors may indicate with arrows or other simpler ways to understand for the readers. Same comments for Figure 2.

We revised Figures 1 and 2 by mainly drawing elliptic arrows indicating rotation angles of the images. We also corrected or added some annotation words in the Results section and Figure legends, correspondingly.

>Figure legends (Fig 1-7) are missing. The authors have only provided the Supplementary figure legends. It’s very hard to evaluate the results without figure legends.

According to the instructions for authors to prepare manuscript, the figure captions and legends should appear directly after the paragraph in which they are first cited. We followed the instruction and did not put the Figure legends section in the initial manuscript. However, the location of Figure legend of Fig2 was not suitable, and there were 2 copies of Figure legend of Fig6 in the initial manuscript, these mistakes are corrected in the Revised Manuscript with Track Changes and the Manuscript.

Other revised points:

The mouse antisera were raised against recombinant domains of chicken ZP1 that were derived from our originally cloned cDNA of chicken ZP1. Therefore, we mentioned accession number of the cDNA in lines 170–172 of the Revised Manuscript with Track Changes and lines 169–171 of the Manuscript.

We had not mention how to assign the ZP3 spots that were separated by LP-IEF and detected in the 2D-PAGE gels to each ZP3 isoforms in the initial manuscript. Therefore, we added the sentence “ZP3 spots detected in the 2D-PAGE gels were assigned to each ZP3 isoforms based on the aforementioned IEF markers and/or comparison of the spot patterns to that of ZP3 isoforms in the F1 IPVL.” in lines 256–257 of the Revised Manuscript with Track Changes and lines 255–256 of the Manuscript.

 Current affiliation of the author, Shunsuke Nishio in line 9, was added in lines 25–26.

---

## [Editor Report · Decision Letter 1]

20 Jul 2022

PONE-D-22-07920R1New Insights into the Role of Microheterogeneity of ZP3 During Structural Maturation of the Avian Equivalent of Mammalian Zona PellucidaPLOS ONE

Dear Dr. Okumura,

Thank you for submitting your manuscript to PLOS ONE. After careful consideration, we feel that it has merit but does not fully meet PLOS ONE’s publication criteria as it currently stands. Therefore, we invite you to submit a revised version of the manuscript that addresses the points raised during the review process. Please consider the following major questions raised by reviewer and Academic Editor:

Major concern:

Please do the English proofreading to avoid any writing errors.As indicated by reviewer, it would be better to writing the sampling procedure clearly

We look forward to receiving your revised manuscript.

Kind regards,

Birendra Mishra, DVM, PhD

Academic Editor

PLOS ONE

Journal Requirements:

Additional Editor Comments:

Authros have adequately responded to the reviewer's comments. However, there are some writing errors in the text of the manuscript, which require English proofreading before the final decision.

It is also not clear about the sampling. The authors mentioned that the blood and ovary were collected immediately. Does it mean that samples were collected immediately before scalding in ~63°C of hot water for 80 seconds? Please make these statements clear.

---

## [Author Response · Author response to Decision Letter 1]

27 Jul 2022

Thank you for reviewing our manuscript entitled “New Insights into the Role of Microheterogeneity of ZP3 During Structural Maturation of the Avian Equivalent of Mammalian Zona Pellucida”. According to reviewers’ comments, we revised our manuscript, and made answers as follows. 

Answers to the question:

> It is also not clear about the sampling. The authors mentioned that the blood and ovary were collected immediately. Does it mean that samples were collected immediately before scalding in ~63°C of hot water for 80 seconds? Please make these statements clear.

 The sentence “The blood and ovary were collected immediately and transported to our laboratory in an ice box.” in lines 130–131 of the Revised Manuscript with Track Changes was replaced with “The blood and ovary were collected during the bleeding and immediately after the removal of feathers, respectively, and transported to our laboratory in an ice box.” in lines 129–131 of the Manuscript. Ovaries of chicken are deeply encased in the body cavity being surrounded by thick adipose tissues, and therefore, denaturation of the egg coat must not be caused by the brief scalding in ~63°C of hot water for 80 seconds.

Other revised points 1) to 17) including corrections of English grammatical errors:

1) The phrase “with approval from the committee” in line 126 of the Revised Manuscript with Track Changes was deleted in line 126 of the Manuscript to correct a grammatical error.

2) The phrase “with approval from the committee” in line 135 of the Revised Manuscript with Track Changes was replaced with “with the approval” in line 135 of the Manuscript to avoid repetition.

3) The phrase “in Figs 1 and Supplement Figs S1” in line 308 of the Revised Manuscript with Track Changes was replaced with “in Fig 1 and S1 Fig” in line 307 of the Manuscript to meet PLOS ONE’s style.

4) The sentence “Proportions of the signal intensities from ZP3 isoforms 3 to 7 are compared between F1 IPVLs of the commercial White-Leghorn and WL-G hens, F1 and F2, F2 and F3, and F1 and F3 IPVLs of the commercial White-Leghorn hens are compared by chi-square test (P-values on the right side of the graphs).” in lines 430–433 of the Revised Manuscript with Track Changes was replaced with “Proportions of the signal intensities from ZP3 isoforms 3 to 7 are compared between F1 IPVLs of the commercial White-Leghorn and WL-G hens, and between F1 and F2, F2 and F3, and F1 and F3 IPVLs of the commercial White-Leghorn hens, respectively, by chi-square test (P-values on the right side of the graphs).” in lines 428–431 of the Manuscript to improve the readability and to correct a grammatical error.

5) The word “were” in line 518 of the Revised Manuscript with Track Changes was replaced with “are” in line 516 of the Manuscript to correct the tense.

6) The phrase “lanes d to f” in line 534 of the Revised Manuscript with Track Changes was replaced with “lanes d–f” in line 532 of the Manuscript to use the same notation.

7) The phrase “already there are distinct differences” in lines 616–617 of the Revised Manuscript with Track Changes was replaced with “there already be distinct differences” in lines 614–615 of the Manuscript to correct a grammatical error.

8) The phrase “and F1 follicles (Fig 2B and S2D Fig)” in line 654 of the Revised Manuscript with Track Changes was replaced with “and F1 (Fig 2B and S2D Fig) follicles” in line 652 of the Manuscript to improve the readability.

9) The phrase “These results suggest that” in lines 659–660 of the Revised Manuscript with Track Changes was replaced with “These observations suggest that” in lines 657–658 of the Manuscript.

10) The phrase “the ZP3 isoforms 3–7” in lines 673–674 of the Revised Manuscript with Track Changes was replaced with “the ZP3 isoforms 3 to 7” in lines 670–671 of the Manuscript to use the same notation.

11) The phrase “these data suggested that the expression levels of individuals ZP3 isoforms” in lines 684–685 of the Revised Manuscript with Track Changes was replaced with “these data suggest that the expression levels of individual ZP3 isoforms” in lines 681–682 of the Manuscript to correct the tense and a grammatical error.

12) The phrase “that are associated with” in line 787 of the Revised Manuscript with Track Changes was replaced with “that might be associated with” in line 784 of the Manuscript.

13) The phrase “in the similar format as Fig 1” in line 925 of the Revised Manuscript with Track Changes was replaced with “in the same format as Fig 1” in line 922 of the Manuscript.

14) The phrase “in both B and C” in line 926 of the Revised Manuscript with Track Changes was replaced with “in both B and C” in line 923 of the Manuscript to correct styles of A and B.

15) The phrase “indicated by white elliptic arrows” in line 926 of the Revised Manuscript with Track Changes was replaced with “(indicated by white elliptic arrows)” in line 923 of the Manuscript to improve the readability using parentheses.

16) The phrase “in the similar format as Fig 2” in line 934 of the Revised Manuscript with Track Changes was replaced with “in the same format as Fig 2” in line 931 of the Manuscript.

17) The phrase “white dotted lines” in line 934 of the Revised Manuscript with Track Changes was replaced with “gray dotted lines” in line 931 of the Manuscript.

---

## [Editor Report · Decision Letter 2]

29 Aug 2022

PONE-D-22-07920R2New Insights into the Role of Microheterogeneity of ZP3 During Structural Maturation of the Avian Equivalent of Mammalian Zona PellucidaPLOS ONE

Dear Dr. Okumura,

Thank you for submitting your manuscript to PLOS ONE. After careful consideration, we feel that it has merit but does not fully meet PLOS ONE’s publication criteria as it currently stands. Therefore, we invite you to submit a revised version of the manuscript that addresses the points raised during the review process.

ACADEMIC EDITOR: 

The authors have responded to all the comments, however, there are several writing errors that I had pointed out in the previous version. I have edited/corrected most of the errors and suggested to chem if any errors (see ATTACHED file). Authors need to check further and confirm if there are any writing errors. After confirming the writing error, the manuscript will be acceptable for publication.

We look forward to receiving your revised manuscript.

Kind regards,

Birendra Mishra, DVM, PhD

Academic Editor

PLOS ONE
---

## [Author Response · Author response to Decision Letter 2]

30 Aug 2022

Thank you for the detailed reviewing our manuscript entitled “New Insights into the Role of Microheterogeneity of ZP3 During Structural Maturation of the Avian Equivalent of Mammalian Zona Pellucida”. According to academic editor’s comments, we confirmed and corrected 144 points of writing errors including your corrections as follows.

Corrections of writing errors:

> The authors have responded to all the comments, however, there are several writing errors that I had pointed out in the previous version. I have edited/corrected most of the errors and suggested to chem if any errors (see ATTACHED file). Authors need to check further and confirm if there are any writing errors. After confirming the writing error, the manuscript will be acceptable for publication.

1) A definite article was inserted into “during structural maturation” in line 41 of the Revised Manuscript with Track Changes to be “during the structural maturation” in line 41 of the Manuscript.

2) A comma was inserted into “ZP1, ZP2, ZP3, ZP4, ZPD and ZPAX” in line 52 of the Revised Manuscript with Track Changes to be “ZP1, ZP2, ZP3, ZP4, ZPD, and ZPAX” in line 52 of the Manuscript.

3) A dash in “the sperm–egg” in line 53 of the Revised Manuscript with Track Changes was replaced with a hyphen to be “the sperm-egg” in line 53 of the Manuscript.

4) A comma was inserted into “interactions including” in line 54 of the Revised Manuscript with Track Changes to be “interactions, including” in line 54 of the Manuscript.

5) A comma was inserted into “mZP1–/–, mZP2–/– and mZP3–/–” in line 57 of the Revised Manuscript with Track Changes to be “mZP1–/–, mZP2–/–, and mZP3–/–” in line 57 of the Manuscript.

6) A definite article and a comma were inserted into “ZP1, ZP2 or ZP3 gene” and the singular form was replaced with a plural form in “ZP1, ZP2 or ZP3 gene” in line 59 of the Revised Manuscript with Track Changes to be “the ZP1, ZP2, or ZP3 genes” in line 59 of the Manuscript.

7) A dash in “the sperm–egg” in line 61 of the Revised Manuscript with Track Changes was replaced with a hyphen to be “the sperm-egg” in line 61 of the Manuscript.

8) A comma was inserted into “fertility including” in line 64 of the Revised Manuscript with Track Changes to be “fertility, including” in line 64 of the Manuscript.

9) The phrase “For elucidating these mechanisms” in lines 67 to 68 of the Revised Manuscript with Track Changes was replaced with “To elucidate these mechanisms” in lines 67 to 68 of the Manuscript.

10) A definite article and a comma were inserted into “characterize structural, biochemical and physiological” in lines 68 to 69 of the Revised Manuscript with Track Changes to be “characterize the structural, biochemical, and physiological” in line 68 of the Manuscript.

11) The phrase “samples that are collected from” in line 70 of the Revised Manuscript with Track Changes was replaced with “samples collected from” in lines 69 to 70 of the Manuscript.

12) A comma was inserted into “poultry species including” in line 75 of the Revised Manuscript with Track Changes to be “poultry species, including” in line 74 of the Manuscript.

13) A comma in “formed quickly,” in line 76 of the Revised Manuscript with Track Changes was deleted to be “formed quickly” in line 76 of the Manuscript.

14) A comma was inserted into “F3, F2 and F1” in line 104 of the Revised Manuscript with Track Changes to be “F3, F2, and F1” in line 104 of the Manuscript.

15) An indefinite article was inserted into “in few days” in line 108 of the Revised Manuscript with Track Changes to be “in a few days” in line 108 of the Manuscript.

16) A definite article was inserted into “process of egg-coat matrix” in line 117 of the Revised Manuscript with Track Changes to be “process of the egg-coat matrix” in line 117 of the Manuscript.

17) A definite article in “during the ovarian follicle” in line 118 of the Revised Manuscript with Track Changes was deleted to be “during ovarian follicle” in line 118 of the Manuscript.

18) A comma was inserted into “slaughtered by hand” in line 128 of the Revised Manuscript with Track Changes to be “slaughtered by hand,” in line 128 of the Manuscript.

19) The phrase “to be remove feathers” in line 129 of the Revised Manuscript with Track Changes was replaced with “to remove feathers” in line 129 of the Manuscript.

20) A definite article was inserted into “guidelines of Meijo University” in line 134 of the Revised Manuscript with Track Changes to be “guidelines of the Meijo University” in line 134 of the Manuscript.

21) A definite article in “with the approval” in line 135 of the Revised Manuscript with Track Changes was deleted to be “with approval” in line 135 of the Manuscript.

22) A comma was inserted into “Respective approximately” in line 143 of the Revised Manuscript with Track Changes to be “Respective, approximately” in line 143 of the Manuscript.

23) A definite article was inserted into “cloned into pGEM-T easy vector” in line 157 of the Revised Manuscript with Track Changes to be “cloned into the pGEM-T easy vector” in line 157 of the Manuscript.

24) The phrase “IPVLs that were isolated from” in line 164 of the Revised Manuscript with Track Changes was replaced with “IPVLs isolated from” in line 164 of the Manuscript.

25) A definite article in “in the non-reducing conditions” in line 192 of the Revised Manuscript with Track Changes was deleted to be “in non-reducing conditions” in line 192 of the Manuscript.

26) A comma was inserted into “(CBB) staining and” in line 194 of the Revised Manuscript with Track Changes to be “(CBB) staining, and” in line 194 of the Manuscript.

27) A comma was inserted into “±0.4, ±0.5, ±1.0 or ±1.1 Da” in line 231 of the Revised Manuscript with Track Changes to be “±0.4, ±0.5, ±1.0, or ±1.1 Da” in line 231 of the Manuscript.

28) The phrase “the rest of fractions” in line 248 of the Revised Manuscript with Track Changes was replaced with “the remaining fractions” in line 248 of the Manuscript.

29) A definite article was inserted into “Examination of binding capacity” in line 258 of the Revised Manuscript with Track Changes to be “Examination of the binding capacity” in line 258 of the Manuscript.

30) A definite article was inserted into “pulled by magnetic field” in line 264 of the Revised Manuscript with Track Changes to be “pulled by the magnetic field” in line 264 of the Manuscript.

31) An indefinite article was inserted into “pulled by magnetic field” in line 266 of the Revised Manuscript with Track Changes to be “pulled by a magnetic field” in line 266 of the Manuscript.

32) A comma was inserted into “(Merck, Darmstadt, Germany) being transformed” in line 278 of the Revised Manuscript with Track Changes to be “(Merck, Darmstadt, Germany), being transformed” in line 278 of the Manuscript.

33) An indefinite article was inserted into “pulled by magnetic field” in line 289 of the Revised Manuscript with Track Changes to be “pulled by a magnetic field” in line 289 of the Manuscript.

34) An indefinite article was inserted into “pulled by magnetic field” in line 291 of the Revised Manuscript with Track Changes to be “pulled by a magnetic field” in line 291 of the Manuscript.

35) An indefinite article was inserted into “pulled by magnetic field” in lines 293 to 294 of the Revised Manuscript with Track Changes to be “pulled by a magnetic field” in lines 293 to 294 of the Manuscript.

36) The phrase “the 3D Viewer plug in” in line 299 of the Revised Manuscript with Track Changes was replaced with “the 3D Viewer plugin” in line 299 of the Manuscript.

37) A comma in “the Sigma settings of 0.001),” in line 303 of the Revised Manuscript with Track Changes was deleted to be “the Sigma settings of 0.001)” in line 303 of the Manuscript.

38) A definite article in “the all analyzed signals” in line 310 of the Revised Manuscript with Track Changes was deleted to be “all analyzed signals” in line 310 of the Manuscript.

39) A definite article in “the Cochran's rule” in line 313 of the Revised Manuscript with Track Changes was deleted to be “Cochran's rule” in lines 312 to 313 of the Manuscript.

40) A comma was inserted into “S2A–S2C Figs) and” in line 324 of the Revised Manuscript with Track Changes to be “S2A–S2C Figs), and” in line 324 of the Manuscript.

41) A comma was inserted into “panels 2, 2–3 or 2–4” in line 329 of the Revised Manuscript with Track Changes to be “panels 2, 2–3, or 2–4” in line 329 of the Manuscript.

42) A comma was inserted into “F2 (B) and F1 (C)” in line 339 of the Revised Manuscript with Track Changes to be “F2 (B), and F1 (C)” in line 329 of the Manuscript.

43) A comma was inserted into “150° and 180°” in line 343 of the Revised Manuscript with Track Changes to be “150°, and 180°” in line 343 of the Manuscript.

44) The phrase “cross section” in line 347 of the Revised Manuscript with Track Changes was replaced with “the cross-section” in line 347 of the Manuscript.

45) A comma was inserted into “the Fig 1 except that” in line 353 of the Revised Manuscript with Track Changes to be “the Fig 1, except that” in line 353 of the Manuscript.

46) The phrase “inside the panels 1” in line 357 of the Revised Manuscript with Track Changes was replaced with “inside panel 1” in line 357 of the Manuscript.

47) A comma in “(panels 2 and 3 in A, and panel 2 in B)” in line 361 of the Revised Manuscript with Track Changes was deleted to be “(panels 2 and 3 in A and panel 2 in B)” in line 361 of the Manuscript.

48) The plural form was replaced with a singular form in “one surfaces of the IPVLs” in line 364 of the Revised Manuscript with Track Changes to be “one surface of the IPVLs” in line 59 of the Manuscript.

49) A comma was inserted into “the F3, F2 and F1 follicles” in line 368 of the Revised Manuscript with Track Changes to be “the F3, F2, and F1 follicles” in line 368 of the Manuscript.

50) A comma was inserted into “~2, ~7 and ~7 µm” in line 368 of the Revised Manuscript with Track Changes to be “~2, ~7, and ~7 µm” in line 368 of the Manuscript.

51) A comma was inserted into “the F3, F2 and F1 follicles” in line 369 of the Revised Manuscript with Track Changes to be “the F3, F2, and F1 follicles” in line 369 of the Manuscript.

52) A comma was inserted into “1.4 ± 0.30 and 1.5 ± 0.58 µm” in lines 369 to 370 of the Revised Manuscript with Track Changes to be “1.4 ± 0.30, and 1.5 ± 0.58 µm” in lines 369 to 370 of the Manuscript.

53) The phrase “similar to each other, a number of holes” in line 371 of the Revised Manuscript with Track Changes was replaced with “similar, several holes” in line 371 of the Manuscript.

54) A comma was inserted into “panels 10 and 11 in Fig 1B, S1B and S1C Figs” in line 373 of the Revised Manuscript with Track Changes to be “panels 10 and 11 in Fig 1B, S1B, and S1C Figs” in line 373 of the Manuscript.

56) A definite article in “cross sections of the panels 9” in line 373 of the Revised Manuscript with Track Changes was deleted to be “cross sections of panels 9” in line 373 of the Manuscript.

57) A comma was inserted into “15°, 0° and 30°” in line 374 of the Revised Manuscript with Track Changes to be “15°, 0°, and 30°” in line 374 of the Manuscript.

58) The phrase “between the surface layers” in line 380 of the Revised Manuscript with Track Changes was replaced with “between their surface layers” in line 380 of the Manuscript.

59) A definite article in “in the panel 1” in line 381 of the Revised Manuscript with Track Changes was deleted to be “in panel 1” in line 381 of the Manuscript.

60) A comma was inserted into “protrusions including” in line 385 of the Revised Manuscript with Track Changes to be “protrusions, including” in line 385 of the Manuscript.

61) A definite article in “including the both surface” in line 385 of the Revised Manuscript with Track Changes was deleted to be “including both surface” in line 385 of the Manuscript.

62) A definite article was inserted into “deeper region” in line 389 of the Revised Manuscript with Track Changes to be “the deeper region” in line 388 of the Manuscript.

63) The phrase “IPVLs that were collected from” in line 412 of the Revised Manuscript with Track Changes was replaced with “IPVLs collected from” in line 412 of the Manuscript.

64) The phrase “proteins that were also detected” in line 416 of the Revised Manuscript with Track Changes was replaced with “proteins also detected” in line 416 of the Manuscript.

65) The phrase “similarly to the panel 1, although ranging from” in line 419 of the Revised Manuscript with Track Changes was replaced with “similarly to panel 1, ranging from” in line 419 of the Manuscript.

66) A definite article in “similarly to the panel 3” in line 421 of the Revised Manuscript with Track Changes was deleted to be “similarly to panel 3” in line 421 of the Manuscript.

67) The phrase “shown on the bottom of the graph d” in line 427 of the Revised Manuscript with Track Changes was replaced with “shown at the bottom of graph d” in lines 426 to 427 of the Manuscript.

68) The phrase “although that were” in line 442 of the Revised Manuscript with Track Changes was replaced with “although they were” in line 441 of the Manuscript.

69) The word “obtained” in line 451 of the Revised Manuscript with Track Changes was deleted in line 450 of the Manuscript.

70) A definite article was inserted into “in S1 Table” in line 462 of the Revised Manuscript with Track Changes to be “in the S1 Table” in line 461 of the Manuscript.

71) The phrase “the peptides that are detected” in line 473 of the Revised Manuscript with Track Changes was replaced with “the peptides detected” in line 472 of the Manuscript.

72) The phrase “residues that contain the identified modifications” in line 474 of the Revised Manuscript with Track Changes was replaced with “residues containing the identified modifications” in line 473 of the Manuscript.

73) The phrase “mutants being derived from” in line 475 of the Revised Manuscript with Track Changes was replaced with “mutants derived from” in line 474 of the Manuscript.

74) A comma was inserted into “318/320–329 and 330–347” in line 486 of the Revised Manuscript with Track Changes to be “318/320–329, and 330–347” in line 485 of the Manuscript.

75) A comma was inserted into “318/320–329 and 330–347” in line 487 of the Revised Manuscript with Track Changes to be “318/320–329, and 330–347” in line 486 of the Manuscript.

76) A comma was inserted into “D33, A36 and A41” in line 494 of the Revised Manuscript with Track Changes to be “D33, A36, and A41” in line 493 of the Manuscript.

77) A comma was inserted into “N339 and the combination of” in line 497 of the Revised Manuscript with Track Changes to be “N339, and the combination of” in line 496 of the Manuscript.

78) The phrase “histidine residues, which majority of them are” in line 511 of the Revised Manuscript with Track Changes was replaced with “histidine residues, most of which are” in line 510 of the Manuscript.

79) The phrase “26 to 29 of histidine residues” in lines 512 to 513 of the Revised Manuscript with Track Changes was replaced with “26 to 29 histidine residues” in line 511 of the Manuscript.

80) A definite article was inserted into “Migration position” in line 533 of the Revised Manuscript with Track Changes to be “The migration position” in line 532 of the Manuscript.

81) The phrase “the fractions that were obtained from” in line 545 of the Revised Manuscript with Track Changes was replaced with “the fractions obtained from” in line 544 of the Manuscript.

82) A comma was inserted into “the fractions 3, 7 and 10” in line 547 of the Revised Manuscript with Track Changes to be “the fractions 3, 7, and 10” in line 546 of the Manuscript.

83) A comma was inserted into “5 to 7 and 7 to 8” in line 548 of the Revised Manuscript with Track Changes to be “5 to 7, and 7 to 8” in line 547 of the Manuscript.

84) The phrase “the fractions that were obtained from” in line 550 of the Revised Manuscript with Track Changes was replaced with “the fractions obtained from” in line 549 of the Manuscript.

85) A comma was inserted into “the fractions 6, 8 and 9” in lines 552 to 553 of the Revised Manuscript with Track Changes to be “the fractions 6, 8, and 9” in line 551 of the Manuscript.

86) A comma was inserted into “isoforms 5, 6 to 7 and 7” in line 553 of the Revised Manuscript with Track Changes to be “isoforms 5, 6 to 7, and 7” in line 552 of the Manuscript.

87) A comma was inserted into “fractions d, e and f” in line 554 of the Revised Manuscript with Track Changes to be “fractions d, e, and f” in line 553 of the Manuscript.

88) A typo “re-fractionaton” in line 566 of the Revised Manuscript with Track Changes was replaced with “re-fractionation in line 564 of the Manuscript.

89) A comma was inserted into “the fractions 3, 7 and 10” in line 567 of the Revised Manuscript with Track Changes to be “the fractions 3, 7, and 10” in line 565 of the Manuscript.

90) A comma was inserted into “fractions a, b and c” in line 568 of the Revised Manuscript with Track Changes to be “fractions a, b, and c” in line 565 of the Manuscript.

91) A comma was inserted into “the fractions 6, 8 and 9” in line 568 of the Revised Manuscript with Track Changes to be “the fractions 6, 8, and 9” in line 566 of the Manuscript.

92) A comma was inserted into “fractions d, e and f” in line 569 of the Revised Manuscript with Track Changes to be “fractions d, e, and f” in lines 566 to 567 of the Manuscript.

93) A definite article was inserted into “the top of bars” in line 578 of the Revised Manuscript with Track Changes to be “the top of the bars” in line 576 of the Manuscript.

94) A comma was inserted into “single, double and triple asterisks” in line 587 of the Revised Manuscript with Track Changes to be “single, double, and triple asterisks” in line 585 of the Manuscript.

95) A comma in “Whereas, that” in line 596 of the Revised Manuscript with Track Changes was deleted to be “Whereas that” in line 594 of the Manuscript.

96) The phrase “Taken together, these results” in lines 598 to 599 of the Revised Manuscript with Track Changes was replaced with “These results” in lines 596 to 597 of the Manuscript.

97) The phrase “isoform 7 by itself exhibited” in line 599 of the Revised Manuscript with Track Changes was replaced with “isoform 7 exhibited” in line 597 of the Manuscript.

98) A definite article in “there might be the possible synergistic effects” in line 600 of the Revised Manuscript with Track Changes was deleted to be “there might be possible synergistic effects” in line 598 of the Manuscript.

99) A comma was inserted into “egg coat including” in line 607 of the Revised Manuscript with Track Changes to be “egg coat, including” in line 604 of the Manuscript.

100) Commas were inserted into “(IPVL) and so on is” in line 608 of the Revised Manuscript with Track Changes to be “(IPVL), and so on, is” in line 605 of the Manuscript.

101) The phrase “showed that there are distinct differences” in line 609 of the Revised Manuscript with Track Changes was replaced with “showed distinct differences” in line 606 of the Manuscript.

102) An indefinite article was inserted into “is composed of more robust” in line 613 of the Revised Manuscript with Track Changes to be “is composed of a more robust” in line 610 of the Manuscript.

103) The phrase “there already be distinct” in line 616 of the Revised Manuscript with Track Changes was replaced with “there are distinct” in line 613 of the Manuscript.

104) An indefinite article was inserted into “one surface has smooth” in line 617 of the Revised Manuscript with Track Changes to be “one surface has a smooth” in line 614 of the Manuscript.

105) A comma was inserted into “panels 1, 2, 5 and 6” in line 618 of the Revised Manuscript with Track Changes to be “panels 1, 2, 5, and 6” in line 615 of the Manuscript.

106) A comma was inserted into “panels 3, 4, 7 and 8” in line 619 of the Revised Manuscript with Track Changes to be “panels 3, 4, 7, and 8” in line 616 of the Manuscript.

107) A definite article was inserted into “formation of egg-coat matrix” in line 620 of the Revised Manuscript with Track Changes to be “formation of the egg-coat matrix” in line 617 of the Manuscript.

108) A comma was inserted into “ZP1, ZP3 and ZPD” in line 625 of the Revised Manuscript with Track Changes to be “ZP1, ZP3, and ZPD” in lines 621 to 622 of the Manuscript.

109) The phrase “interestingly, there are some cavities and tunnels traversing” in lines 633 to 634 of the Revised Manuscript with Track Changes was replaced with “interestingly, some cavities and tunnels are traversing” in lines 630 to 631 of the Manuscript.

110) A comma was inserted into “Fig 1B, S1B, S1C and S1E Figs” in line 644 of the Revised Manuscript with Track Changes to be “Fig 1B, S1B, S1C, and S1E Figs” in line 641 of the Manuscript.

111) A comma was inserted into “the F2 IPVLs suggest that” in line 648 of the Revised Manuscript with Track Changes to be “the F2 IPVLs, suggest that” in line 645 of the Manuscript.

112) A comma was inserted into “panels 3, 4, 7 and 8” in line 659 of the Revised Manuscript with Track Changes to be “panels 3, 4, 7, and 8” in line 656 of the Manuscript.

113) Commas were inserted into “proteins including chicken ZP3 are” in line 669 of the Revised Manuscript with Track Changes to be “proteins, including chicken ZP3, are” in line 666 of the Manuscript.

114) The phrase “Of note, the relative proportions” in line 672 of the Revised Manuscript with Track Changes was replaced with “Notably, the relative proportions” in line 669 of the Manuscript.

115) The phrase “In our previous study, it is suggested that” in line 689 of the Revised Manuscript with Track Changes was replaced with “Our previous study suggests that” in line 669 of the Manuscript.

116) A comma was inserted into “some modifications including” in line 692 of the Revised Manuscript with Track Changes to be “some modifications, including” in line 689 of the Manuscript.

117) A comma was inserted into “partially conserved at least” in lines 706 to 707 of the Revised Manuscript with Track Changes to be “partially conserved, at least” in line 703 of the Manuscript.

118) The phrase “chicken IPVLs, while the chicken line-specific” in line 708 of the Revised Manuscript with Track Changes was replaced with “chicken IPVLs. In contrast, the chicken line-specific” in lines 704 to 705 of the Manuscript.

119) The phrase “each chicken line, although little is known” in line 709 of the Revised Manuscript with Track Changes was replaced with “each chicken line. However, little is known” in lines 705 to 706 of the Manuscript.

120) A definite article was inserted into “regulating physiological” in line 710 of the Revised Manuscript with Track Changes to be “regulating the physiological” in line 706 of the Manuscript.

121) A comma was inserted into “isoforms 3, 4 and 7” in line 711 of the Revised Manuscript with Track Changes to be “isoforms 3, 4, and 7” in lines 707 to 708 of the Manuscript.

122) A comma was inserted into “isoforms 3, 4 and 7” in line 715 of the Revised Manuscript with Track Changes to be “isoforms 3, 4, and 7” in line 711 of the Manuscript.

123) A definite article was inserted into “Key findings of” in lines 759 to760 of the Revised Manuscript with Track Changes to be “The key findings of” in lines 755 to 756 of the Manuscript.

124) The phrase “isoform 7 by itself exhibits” in line 776 of the Revised Manuscript with Track Changes was replaced with “isoform 7 exhibits” in line 772 of the Manuscript.

125) A comma was inserted into “isoforms 3 to 7 at least in” in line 777 of the Revised Manuscript with Track Changes to be “isoforms 3 to 7, at least in” in line 773 of the Manuscript.

126) A comma was inserted into “granulosa cells although” in line 784 of the Revised Manuscript with Track Changes to be “granulosa cells, although” in line 780 of the Manuscript.

127) A comma was inserted into “F3 (A), F2 (B, C and E) and F1 (G) IPVLs” in lines 925 to 926 of the Revised Manuscript with Track Changes to be “F3 (A), F2 (B, C and E), and F1 (G) IPVLs” in lines 921 to 922 of the Manuscript.

128) The singular form was replaced with a plural form in “structure on IPVLs” in line 934 of the Revised Manuscript with Track Changes to be “structures on IPVLs” in line 930 of the Manuscript.

129) The phrase “in the same format as” in line 935 of the Revised Manuscript with Track Changes was replaced with “in the similar format as” in line 931 of the Manuscript.

130) The phrase “although all ranging from” in line 947 of the Revised Manuscript with Track Changes was replaced with “although all are ranging from” in line 943 of the Manuscript.

131) A definite article was inserted into “and positions of” in line 950 of the Revised Manuscript with Track Changes to be “and the positions of” in line 946 of the Manuscript.

132) A definite article in “immunoblot images of the Fig 5” in line 954 of the Revised Manuscript with Track Changes was deleted to be “immunoblot images of Fig 5” in line 950 of the Manuscript.

133) A comma was inserted into “fractions and” in line 955 of the Revised Manuscript with Track Changes to be “fractions, and” in line 951 of the Manuscript.

134) The phrase “fractions that were collected from” in line 958 of the Revised Manuscript with Track Changes was replaced with “fractions collected from” in line 954 of the Manuscript.

135) A definite article was inserted into “top of images” in line 960 of the Revised Manuscript with Track Changes to be “top of the images” in line 956 of the Manuscript.

136) The phrase “fractions 5 and 6 that are seen in A” in line 964 of the Revised Manuscript with Track Changes was replaced with “fractions 5 and 6 seen in A” in lines 959 to 960 of the Manuscript.

137) A definite article in “gel images of the Figs 6A” in line 965 of the Revised Manuscript with Track Changes was deleted to be “gel images of Figs 6A” in line 961 of the Manuscript.

138) A comma was inserted into “ZP3 and the MW markers” in line 966 of the Revised Manuscript with Track Changes to be “ZP3, and the MW markers” in line 962 of the Manuscript.

139) A comma was inserted into “ZP3 and the MW markers” in line 974 of the Revised Manuscript with Track Changes to be “ZP3, and the MW markers” in line 970 of the Manuscript.

140) A definite article in “blot images of the Figs 6C” in line 975 of the Revised Manuscript with Track Changes was deleted to be “blot images of Figs 6C” in line 971 of the Manuscript.

141) A comma was inserted into “ZP3 and the MW markers” in line 977 of the Revised Manuscript with Track Changes to be “ZP3, and the MW markers” in line 973 of the Manuscript.

142) A comma was inserted into “The single, double and triple” in line 977 of the Revised Manuscript with Track Changes to be “The single, double, and triple” in line 973 of the Manuscript.

143) The phrase “sequences being identified with” in line 981 of the Revised Manuscript with Track Changes was replaced with “sequences identified with” in line 977 of the Manuscript.

144) The phrase “peptides are shown by” in line 982 of the Revised Manuscript with Track Changes was replaced with “peptides are indicated by” in line 978 of the Manuscript.

---

## [Decision Letter · Decision Letter 3]

3 Nov 2022

PONE-D-22-07920R3New Insights into the Role of Microheterogeneity of ZP3 During Structural Maturation of the Avian Equivalent of Mammalian Zona PellucidaPLOS ONE

Dear Dr. Okumura,

Thank you for submitting your manuscript to PLOS ONE. After careful consideration, we feel that it has merit but does not fully meet PLOS ONE’s publication criteria as it currently stands. Therefore, we invite you to submit a revised version of the manuscript that addresses the points raised during the review process.

ACADEMIC EDITOR: Please respond to the questions raised by the reviewers and also check for any additional writing errors.

We look forward to receiving your revised manuscript.

Kind regards,

Birendra Mishra, DVM, PhD

Academic Editor

PLOS ONE

Journal Requirements:

Additional Editor Comments (if provided):

Authors have responded to my comments/suggestions. Thanks

Reviewers' comments:

Reviewer's Responses to Questions

**Comments to the Author**

1. If the authors have adequately addressed your comments raised in a previous round of review and you feel that this manuscript is now acceptable for publication, you may indicate that here to bypass the “Comments to the Author” section, enter your conflict of interest statement in the “Confidential to Editor” section, and submit your "Accept" recommendation.

Reviewer #1: (No Response)

Reviewer #2: (No Response)

2. Is the manuscript technically sound, and do the data support the conclusions?

Reviewer #1: Yes

Reviewer #2: Partly

3. Has the statistical analysis been performed appropriately and rigorously? 

Reviewer #1: Yes

Reviewer #2: I Don't Know

4. Have the authors made all data underlying the findings in their manuscript fully available?

Reviewer #1: Yes

Reviewer #2: Yes

5. Is the manuscript presented in an intelligible fashion and written in standard English?

Reviewer #1: Yes

Reviewer #2: No

6. Review Comments to the Author

Reviewer #1: This research group/lab continues to provide insightful and comprehensive research regarding zona pellucida proteins, especially concerning the avian zona pellucida. The reviewed manuscript fits with the quality that is known with this research group. The research is the most comprehensive study to look at the molecular mechanism concerning the assembly of the zona pellucida (or IPVL). The manuscript follows up on a previous report concerning microheterogenicity of ZPC in the IPVL. However, the current report uses several techniques, some unique such as the creation of a quantifiable ZP3-ZP1 binding sassy. The comprehensive battery of test provides sufficient evidence that supports the authors conclusions and insights that are summarized well in Figure 7. Overall, I do not find any errors in methodology that would prevent this from being published.

The manuscript is well written and there are no glaring grammatical errors of note:

Only issues worth pointing out:

The reason using two different lines WL-G and commercial white leghorns was not clearly communicated. For that matter, the route of euthanasia for the WL-G lines was also not stated.

Lines 206-223: Was only one sample per ZP3 spot submitted for LC MADI MS/MS? If not, how many were submitted and was there any error among replicate sequences?

Reviewer #2: The manuscript by Okomura and colleagues investigates the role of ZP3 microheterogeneity during maturation of the inner perivitelline layer (IPVL) of the hen egg at the end of rapid follicular growth. IPVL is a structure consisting of protein fibers that cover the oocyte. The mechanisms of its formation in the late stages of follicular maturation is poorly understood until now, and is associated with a key question: how does the formation of this fibrous layer adapt to the high growth rate of the follicles in the last days before ovulation? The size of the follicles indeed increases by about 1 to 4 cm in the last 5-6 days of maturation before ovulation.

Okumura's study focused on the last 3 days of follicle maturation (F3, F2 and F1). They first studied the 3D structure of the IPVL using a confocal microscopy approach to observe the orientation and size of protein fibers at these different stages. A 2D-SDS-PAGE study revealed the existence of different isoforms of the ZP3 protein (which is one of the major proteins of the IPVL) whose proportions seem to vary during the last days of follicular maturation. Post-translational modifications were revealed by mass spectrometry for each ZP3 spot identified. Finally, they attempted to demonstrate different affinities towards ZP1 for each ZP3 isoform.

This work, although very interesting, lacks clarity and/or precision (especially regarding the methodology and results) and some results (in particular those concerning the ZP1 binding assay) are too preliminary. Some interpretation and conclusions are also questionable, which calls into question the predictive model presented at the end of the manuscript. Consequently, the manuscript needs to be corrected and completed in depth. Substantial modifications are required.

Major comments:

1. The lack of reliable information on the orientation of the forming IPVL in the microscopy study is a major issue. The figures and results do not indicate whether the surfaces with 'smooth or homogeneously granular microstructures' (Fig. 1, panels 1, 2, 5, 6) and 'fibrous microstructures' (Fig. 1, panels 3, 4, 7, 8) are oriented towards the granulosa cells or towards the oocyte. Based on the presence of debris, the authors concluded that the "smooth surface of IPVL may be adjacent to the granulosa cell layer in the ovarian follicles" (line 637-638). I assume that the authors think that these debris come from the granulosa cells. But such debris could also be traces of yolk or remnants of the plasma membrane of the oocyte. The orientation needs to be defined more precisely and unambiguously. Is it possible to identify the orientation of the IPVL during the isolation step? This is very important to clarify because the predictive model for the structural maturation of the IPVL discussed in the manuscript strongly relies on the appearance of the surfaces and their orientation in the follicle. Moreover, it should be noted that some results in the literature tend to show the opposite. Indeed, in the fresh egg, the outer surface of the IPVL exposed after separation from the outer layer (i.e. the surface that was theoretically interacting with the granulosa cells) has thick fibers whereas the surface in contact with the yolk content or the oocyte has smaller fibers (In Kido and Doi, 1988, see Figure 4, panels C vs E, https://doi.org/10.3382/ps.0670476). In this case, the surface with the 'fibrous microstructures' could therefore be adjacent to the granulosa cells.

2. The methodology used for the ZP1 binding assay raises many questions impairing the reliability of the results:

-The ZP1 binding test does not appear to be specific. The ZP3 crude extracts used are not pure and all bands present, including non-ZP3 bands, are found in the eluted fractions (see silver-stained gel, Fig. S5D).

-The protein amounts/concentrations of ZP1 solution and ZP3 crude extracts used are not indicated in the material and method. To compare the binding capacity of different ZP3 fractions, ZP1 beads should normally be incubated with identical amounts of ZP3 protein, but immunoblot results suggest that the starting amounts are different for the different crude extracts used (Figs. 6C and 6G). To overcome this, the authors attempted to normalize their results with the ratio of the starting amount to the eluted amount of ZP3, but this method is not ideal. The presence of more or less abundant contaminating bands or proteins in the ZP3 fractions is problematic. Figures S5B and S5D show that ZP3 band is minor in fraction 10 (crude extract “c”). This fraction even seems to contain contaminating ZP1 protein, which can potentially bind ZP3 protein and prevent it from binding to the ZP1 beads.

-The binding assay is based on a quantification of bands obtained by immunoblot in order to identify the isoforms that best bind the ZP1 protein. As the fractions used contain several isoforms, do the authors know if all isoforms are equally recognised by the antibody? Also, the densitometry results sometimes seem to give inconsistent results. For example, in Figure 6C, band “b” in the crude extract appears visually much more intense than bands “a” and “c”, yet the densitometry values do not agree with this observation. In Figure 6G, band e in the crude extract appears much more intense (at least twice more intense) than bands “d” and “f”, but the densitometry values provided are close for all three bands. In the middle panel (elution), band “e” appears to be more intense than the band “f”, but here the densitometry results are the opposite.

-There is no information on the number of repeats. Was this test repeated several times?

-The authors conclude “these results showed that the ZP3 isoform 7 exhibited much higher binding affinity among ZP3 isoforms 2-8, although there might be possible synergistic effects of these ZP3 isoforms” (line 597-598). The data do not allow to be so affirmative about isoform 7 because fractions “c” and “f”, which according to the authors have the best binding capacity towards ZP1, also contain isoform 6. Why would isoform 7 bind ZP1 better than isoform 6? Furthermore, isoform 7 also appears to be a major component of fraction “e”. It is therefore difficult to conclude with certainty about the isoform that binds ZP1 best.

For all these reasons, the results and conclusions of the ZP1-ZP3 binding test appear preliminary and need to be confirmed, either by using (more) pure ZP3 fractions or by using other methodological approaches.

Additional comments:

1. The study was carried out on White Leghorn hens from two different origins, namely “commercial White Leghorn” and “long-term closed colony White Leghorn”. Why were two types of animals used in this study? In figure 3, the immunoblots from the “commercial White Leghorn” samples are compared with a coomassie gel from “long-term closed colony White Leghorn” animals. But a silver nitrate gel on the “commercial White Leghorn” is also shown in this figure. Why did the authors choose to compare densitometry results from samples of different origins? In panel B, it would be more rigorous to compare spots from samples from the same animals (e.g. the spots from immunoblots with the spots from the silver stained gel). Furthermore, the statistical analysis in Figure 3B does not mention the number of samples or animals used, nor the number of technical replicates.

2. The IPVL shown in Figures 1B, 1C, S1B, S1C, S1E, S1G (panels 2, 3, 6, 7) appear to have two sublayers. This two-layered structure is quite troubling. Is it an artefact or something real? This is not discussed in the results section.

3. Lines 639-648: This part of discussion could be discussed with regards to the ultrastructural observations of the chicken zona radiata made by Wyburn in 1965: Wyburn GM, Aitken RN, Johnston HS. The ultrastructure of the zona radiata of the ovarian follicle of the domestic fowl. J Anat. 1965 Jul;99(Pt 3):469-84.

4. The text needs to be proofread by a native English reader.

Minor remarks:

-Figure 1: Scale values in µm are shown in the lower panels, but not in the upper panels. Please homogenise.

-Figure 4: the asterisk is not described in the figure legend.

-Line 133: the abbreviation MEXT needs to be defined

-Line 211: "1% acrylamide" is probably not the right reagent for alkylation.

-Line 284: the use of "wt%" is not clear

7. PLOS authors have the option to publish the peer review history of their article (what does this mean?). If published, this will include your full peer review and any attached files.

Reviewer #1: No

Reviewer #2: No

---

## [Author Response · Author response to Decision Letter 3]

7 Nov 2022

Thank you for the detailed reviewing our manuscript entitled “New Insights into the Role of Microheterogeneity of ZP3 During Structural Maturation of the Avian Equivalent of Mammalian Zona Pellucida”. According to the academic editor’s and reviewers’ comments, we answered and corrected some points as follows.

• For the comments from the reviewer #1:

Comment 1 from the reviewer #1

> The reason using two different lines WL-G and commercial white leghorns was not clearly communicated. For that matter, the route of euthanasia for the WL-G lines was also not stated.

(Answer)

We used the WL-G (the long-term closed colony of White-Leghorn breed) hens in addition to the commercial White-Leghorn hens to demonstrate that the isoelectric-point microheterogeneity of chicken ZP3 is not due to the potential genetic heterogeneities in commercial hens. In general, chicken especially the commercial hens possess genetic heterogeneities, although we can prepare abundant IPVL from their large ovarian follicles. That is mentioned in lines 676 to 678 of both the Revised Manuscript with Track Changes and the Manuscript, although not clearly described.

To make the reason using the WL-G line and the route of euthanasia for the WL-G lines, we corrected the sentence “Fresh ovary and genomic DNA of the long-term closed colony of White-Leghorn breed (WL-G) hens were provided” was corrected into “To analyze the genetically homogeneous line of hen, fresh ovary and genomic DNA of the long-term closed colony of White-Leghorn breed (WL-G) hens that were killed similarly by hands were provided” in lines 131 to 133 of the Revised Manuscript with Track Changes and lines 131 to 132 of the Manuscript.

Comment 2 from the reviewer #1

> Lines 206-223: Was only one sample per ZP3 spot submitted for LC MADI MS/MS? If not, how many were submitted and was there any error among replicate sequences?

(Answer)

 One sample per ZP3 spot was submitted for LC MALDI MS/MS. However, each spot of all the ZP3 isoforms 3 to 7 were obtained from both the F1 IPVL of commercial White-Leghorn and WL-G hens, and there are some similarities in the AA substitutions and modifications as mentioned in the manuscript.

• For the comments from the reviewer #2:

Comment 1 from the reviewer #2

> 1. The lack of reliable information on the orientation of the forming IPVL in the microscopy study is a major issue. The figures and results do not indicate whether the surfaces with 'smooth or homogeneously granular microstructures' (Fig. 1, panels 1, 2, 5, 6) and 'fibrous microstructures' (Fig. 1, panels 3, 4, 7, 8) are oriented towards the granulosa cells or towards the oocyte. Based on the presence of debris, the authors concluded that the "smooth surface of IPVL may be adjacent to the granulosa cell layer in the ovarian follicles" (line 637-638). I assume that the authors think that these debris come from the granulosa cells. But such debris could also be traces of yolk or remnants of the plasma membrane of the oocyte. The orientation needs to be defined more precisely and unambiguously. Is it possible to identify the orientation of the IPVL during the isolation step? This is very important to clarify because the predictive model for the structural maturation of the IPVL discussed in the manuscript strongly relies on the appearance of the surfaces and their orientation in the follicle. Moreover, it should be noted that some results in the literature tend to show the opposite. Indeed, in the fresh egg, the outer surface of the IPVL exposed after separation from the outer layer (i.e. the surface that was theoretically interacting with the granulosa cells) has thick fibers whereas the surface in contact with the yolk content or the oocyte has smaller fibers (In Kido and Doi, 1988, see Figure 4, panels C vs E, https://doi.org/10.3382/ps.0670476). In this case, the surface with the 'fibrous microstructures' could therefore be adjacent to the granulosa cells.

(Answer)

 As the reviewer #2 pointed out, the presence of debris on the surface of IPVL with “smooth or homogeneously granular microstructures” will be not enough to indicate that this surface was adjacent to the granulosa cell layer in the ovarian follicles. Therefore, the phrase “, although further analyses to determine the orientation of IPVL on the oocyte are required” was added after the sentence “These results imply that the smooth surface of IPVL may be adjacent to the granulosa cell layer in the ovarian follicles.” in lines 641 to 642 of both the Revised Manuscript with Track Changes and the Manuscript.

 In this study, we obtained the surface microstructure images of the IPVL from ovarian follicles under wet conditions as mentioned in the Materials and Method section. Therefore, our results are not directly comparable to the results of electron microscopic studies of IPVL. Furthermore, in the study the reviewer #2 cited, the specimen of IPVL were prepared from vitelline envelope of the fresh egg after the removal of the outer layer by SDS treatment, and the surface microstructure of the IPVL may be considerably denatured.

Comment 2 from the reviewer #2

> 2. The methodology used for the ZP1 binding assay raises many questions impairing the reliability of the results:

-The ZP1 binding test does not appear to be specific. The ZP3 crude extracts used are not pure and all bands present, including non-ZP3 bands, are found in the eluted fractions (see silver-stained gel, Fig. S5D).

(Answer)

 Although the non-ZP3 proteins that may interact with the ZP1 beads are contained in the ZP3 crude extracts, the ZP1–ZP3 binding assay showed that there certainly be significant differences among the binding affinities of ZP3 isoforms against the ZP1 beads.

However, the phrase “, although the non-ZP3 contaminants in the crude ZP3-isoform fractions a–c (see Fig S5D) might interact with the ZP1-beads” was added after the sentence “The relative ratios of the ZP1 binding affinities of ZP3 in the fractions a–c (Fig 6D) showed that ZP3 in the mixture of ZP3 isoforms 5–7 (major components in the fraction b) bound to ZP1 with approximately 3-times higher affinity than ZP3 in the mixtures of ZP3 isoforms 2–4 and 7–8 (major components in the fractions a and c, respectively).” in lines 591 to 596 of both the Revised Manuscript with Track Changes and the Manuscript.

-The protein amounts/concentrations of ZP1 solution and ZP3 crude extracts used are not indicated in the material and method. To compare the binding capacity of different ZP3 fractions, ZP1 beads should normally be incubated with identical amounts of ZP3 protein, but immunoblot results suggest that the starting amounts are different for the different crude extracts used (Figs. 6C and 6G). To overcome this, the authors attempted to normalize their results with the ratio of the starting amount to the eluted amount of ZP3, but this method is not ideal. The presence of more or less abundant contaminating bands or proteins in the ZP3 fractions is problematic. Figures S5B and S5D show that ZP3 band is minor in fraction 10 (crude extract “c”). This fraction even seems to contain contaminating ZP1 protein, which can potentially bind ZP3 protein and prevent it from binding to the ZP1 beads.

(Answer)

 In this ZP1–ZP3 binding assay, we compared ratios of the amount of ZP3 being eluted from the ZP1 beads against the starting amount of ZP3 for each ZP3 crude extract, using incompletely purified ZP1 and ZP3 samples. Therefore, we did not determine the precise amounts of ZP1 and ZP3 in the crude samples, while the differences among the starting amounts of ZP3 are normalized in the calculation of the ratios. To clarify that, the phrase “in the elution fractions to the crude ZP3-isoform fractions” was replaced with “in the elution fractions against the crude ZP3-isoform fractions” in lines 589 to 590 of both the Revised Manuscript with Track Changes and the Manuscript.

 It is still unclear how ZP3 bind to ZP1 (i.e., structural details of ZP1–ZP3 interactions), and we cannot discuss how the contaminating ZP1 affect the ZP3 binding to the ZP1 beads.

-The binding assay is based on a quantification of bands obtained by immunoblot in order to identify the isoforms that best bind the ZP1 protein. As the fractions used contain several isoforms, do the authors know if all isoforms are equally recognised by the antibody? Also, the densitometry results sometimes seem to give inconsistent results. For example, in Figure 6C, band “b” in the crude extract appears visually much more intense than bands “a” and “c”, yet the densitometry values do not agree with this observation. In Figure 6G, band e in the crude extract appears much more intense (at least twice more intense) than bands “d” and “f”, but the densitometry values provided are close for all three bands. In the middle panel (elution), band “e” appears to be more intense than the band “f”, but here the densitometry results are the opposite.

(Answer)

All ZP3 isoforms must be equally recognized by the anti-ZP3 antibody being used in this study, because the antibody is a polyclonal one raised against the recombinant polypeptide of the ZP-C domain of chicken ZP3 that was produced using an E. coli system. Actually, the 2D-PAGE-separated ZP3 isoforms of F1 IPVLs from WL-G and commercial White-Leghorn hens were detected by CBB staining and immunoblotting using the anti-ZP3, respectively, without significant differences in signal intensity patterns (Figs 3A and 3B).

Densitometric values are calculated based on the sum of pixel intensities in the area of gel or blot images of protein bands or spots. Therefore, it sometimes happens that the results of densitometric analyses are inconsistent with the visual appearances of bands or spots, although the densitometric values are affected how to define the area of the signals. We performed the densitometric analyses against figures that the reviewer #2 pointed out, but there are not significant differences influencing the outcome between the re-analyzed results and the original ones used in the manuscript.

-There is no information on the number of repeats. Was this test repeated several times?

(Answer)

 We did not repeat this test. However, we performed several incomplete versions of this test in addition to the more preliminary experiments based on far-western analyses against 2D-PAGE-separated ZP3 isoforms, and the results of them were fit well to the result of this test in summary.

-The authors conclude “these results showed that the ZP3 isoform 7 exhibited much higher binding affinity among ZP3 isoforms 2-8, although there might be possible synergistic effects of these ZP3 isoforms” (line 597-598). The data do not allow to be so affirmative about isoform 7 because fractions “c” and “f”, which according to the authors have the best binding capacity towards ZP1, also contain isoform 6. Why would isoform 7 bind ZP1 better than isoform 6? Furthermore, isoform 7 also appears to be a major component of fraction “e”. It is therefore difficult to conclude with certainty about the isoform that binds ZP1 best.

(Answer)

The phrase “(e.g., the ZP1 binding activity of the isoform 7 might be inhibited by the isoform 6 and be enhanced by the isoform 5)” was added after the sentence “These results showed that the ZP3 isoform 7 exhibited much higher ZP1 binding affinity among ZP3 isoforms 2–8, although there might be possible synergistic effects of these ZP3 isoforms.” in lines 599 to 601 of both the Revised Manuscript with Track Changes and the Manuscript.

For all these reasons, the results and conclusions of the ZP1-ZP3 binding test appear preliminary and need to be confirmed, either by using (more) pure ZP3 fractions or by using other methodological approaches.

(Answer)

 The phrase “, and we need to confirm the results by using pure ZP3 isoform and ZP1 preparations” was added after the sentence “These results showed that the ZP3 isoform 7 exhibited much higher ZP1 binding affinity among ZP3 isoforms 2–8, although there might be possible synergistic effects of these ZP3 isoforms.” in lines 736 to 739 of both the Revised Manuscript with Track Changes and the Manuscript.

Additional comments from the reviewer #2:

1. The study was carried out on White Leghorn hens from two different origins, namely “commercial White Leghorn” and “long-term closed colony White Leghorn”. Why were two types of animals used in this study? In figure 3, the immunoblots from the “commercial White Leghorn” samples are compared with a coomassie gel from “long-term closed colony White Leghorn” animals. But a silver nitrate gel on the “commercial White Leghorn” is also shown in this figure. Why did the authors choose to compare densitometry results from samples of different origins? In panel B, it would be more rigorous to compare spots from samples from the same animals (e.g., the spots from immunoblots with the spots from the silver stained gel). Furthermore, the statistical analysis in Figure 3B does not mention the number of samples or animals used, nor the number of technical replicates.

(Answer)

We used the WL-G (the long-term closed colony of White-Leghorn breed) hens in addition to the commercial White-Leghorn hens to demonstrate that the isoelectric-point microheterogeneity of chicken ZP3 is not due to the potential genetic heterogeneities in commercial hens. In general, chicken especially the commercial hens possess genetic heterogeneities, although we can prepare abundant IPVL from their large ovarian follicles. That is mentioned in lines 676 to 678 of both the Revised Manuscript with Track Changes and the Manuscript, although not clearly described.

To make the reason using the WL-G line and the route of euthanasia for the WL-G lines, we corrected the sentence “Fresh ovary and genomic DNA of the long-term closed colony of White-Leghorn breed (WL-G) hens were provided” was corrected into “To analyze the genetically homogeneous line of hen, fresh ovary and genomic DNA of the long-term closed colony of White-Leghorn breed (WL-G) hens that were killed similarly by hands were provided” in lines 131 to 133 of the Revised Manuscript with Track Changes and lines 131 to 132 of the Manuscript.

2. The IPVL shown in Figures 1B, 1C, S1B, S1C, S1E, S1G (panels 2, 3, 6, 7) appear to have two sublayers. This two-layered structure is quite troubling. Is it an artefact or something real? This is not discussed in the results section.

(Answer)

 The two-layered staining of the IPVL must be an artefact because of the low permeability of the IPVL against macromolecules such as antibodies. We therefore stained both surfaces of IPVL in the immunofluorescent staining (see Materials and Methods). Accordingly, we inserted the phrase “showing much lower permeability against macromolecules including antibodies than F2 IPVLs” in line 651 of both the Revised Manuscript with Track Changes and the Manuscript.

3. Lines 639-648: This part of discussion could be discussed with regards to the ultrastructural observations of the chicken zona radiata made by Wyburn in 1965: Wyburn GM, Aitken RN, Johnston HS. The ultrastructure of the zona radiata of the ovarian follicle of the domestic fowl. J Anat. 1965 Jul;99(Pt 3):469-84.

(Answer)

 The phrase “, as partly mentioned in [14]” was added after the sentence “These results together with the observation that the F1 IPVLs are harder and easier to be detached from the ovarian follicles than the F2 IPVLs, suggest that the loose F2 IPVLs mature quickly into the F1 IPVLs in which the fibrous structures form tight associations with each other showing low permeability against macromolecules including antibodies, and the avian equivalent of mammalian TZPs that may traverse the F2 IPVL are closed before the ovulation in the F1 IPVLs.” in lines 648 to 653 of both the Revised Manuscript with Track Changes and the Manuscript.

4. The text needs to be proofread by a native English reader.

(Answer)

 The reviewer #1 have read our last revised manuscript and answered “Yes” for the question “5. Is the manuscript presented in an intelligible fashion and written in standard English?” from the editor. Therefore, we think our manuscript has become readable enough for any English readers around the world through the last 3-times revisions.

Minor remarks from the reviewer #2:

-Figure 1: Scale values in µm are shown in the lower panels, but not in the upper panels. Please homogenise.

(Answer)

 The scale values in µm are added in the upper panels in Figure 1 and Supplement figure 1.

-Figure 4: the asterisk is not described in the figure legend.

(Answer)

 The asterisk in Figure 4 has already be shown in the figure legend in line 477 of both the Revised Manuscript with Track Changes and the Manuscript as “(*: P<0.01)”.

-Line 133: the abbreviation MEXT needs to be defined

(Answer)

 The abbreviation MEXT was defined as “the Ministry of Education, Culture, Sports, Science and Technology (MEXT)” in line 134 of both the Revised Manuscript with Track Changes and the Manuscript.

-Line 211: "1% acrylamide" is probably not the right reagent for alkylation.

(Answer)

 1% acrylamide are one of the reagents to be used for the cysteine alkylation. See the link below

https://pubmed.ncbi.nlm.nih.gov/17484107/

-Line 284: the use of "wt%" is not clear

(Answer)

 The phrase “30 µl of the 75–80 wt% suspension” was replaced with “30 µl of suspension containing 75–80 wt% beads” in lines 285 to 286 of the Revised Manuscript with Track Changes and line 285 of the Manuscript.

Corrections in the Reference list:

 Reference 15 was corrected with “Okumura H. Avian Egg and Egg Coat. In: Sasanami T, editor. Avian Reproduction: From Behavior to Molecules. Advances in Experimental Medicine and Biology. 1001. Singapore: Springer Singapore; 2017. p. 75-90.” in lines 838 to 840 of both the Manuscript with Track Changes and the Manuscript.

 Reference 16 was corrected with “Nishio S, Okumura H, Matsuda T. Egg-Coat and Zona Pellucida Proteins of Chicken as a Typical Species of Aves. In: Litscher ES and Wassarman PM, editor. Extracellular Matrix and Egg Coats. Current Topics in Developmental Biology. 130. 20180326 ed: Elsevier; 2018. p. 307-29.” in lines 841 to 843 of both the Manuscript with Track Changes and the Manuscript.

 Reference 26 was corrected with “D'Souza CA, Moscarello MA. Differences in susceptibility of MBP charge isomers to digestion by stromelysin-1 (MMP-3) and release of an immunodominant epitope. Neurochemical Research. 2006;31:1045–54.” in lines 876 to 878 of both the Revised Manuscript with Track Changes and the Manuscript.

 Reference 35 was corrected with “Familiari G, Nottola SA, Macchiarelli G, Micara G, Aragona C, Motta PM. Human zona pellucida during in vitro fertilization: An ultrastructural study using saponin, ruthenium red, and osmium-thiocarbohydrazide. Molecular reproduction and development. 1992;32(1):51-61. doi: 10.1002/mrd.1080320109.” in lines 899 to 902 of both the Revised Manuscript with Track Changes and the Manuscript.

---

## [Editor Report · Decision Letter 4]

2 Mar 2023

New Insights into the Role of Microheterogeneity of ZP3 During Structural Maturation of the Avian Equivalent of Mammalian Zona Pellucida

PONE-D-22-07920R4

Dear Dr. Okumura,

We’re pleased to inform you that your manuscript has been judged scientifically suitable for publication and will be formally accepted for publication once it meets all outstanding technical requirements.

Kind regards,

Birendra Mishra, DVM, PhD

Academic Editor

PLOS ONE
---

## [Editor Report · Acceptance letter]

9 Mar 2023

PONE-D-22-07920R4 

New Insights into the Role of Microheterogeneity of ZP3 During Structural Maturation of the Avian Equivalent of Mammalian Zona Pellucida 

Dear Dr. Okumura:

I'm pleased to inform you that your manuscript has been deemed suitable for publication in PLOS ONE. Congratulations! Your manuscript is now with our production department. 

Kind regards, 

on behalf of

Dr. Birendra Mishra 

Academic Editor

PLOS ONE